# Pursuing Feature Separation based on Neural Collapse for Out-of-Distribution Detection

**Yingwen Wu, Ruiji Yu, Xinwen Cheng, Zhengbao He, Xiaolin Huang**[*]
Institute of Image Processing and Pattern Recognition, Shanghai Jiao Tong University
{yingwen_wu, yj1938, xinwencheng, lstefanie, xiaolinhuang}@sjtu.edu.cn

## Abstract

In the open world, detecting out-of-distribution (OOD) data, whose labels are disjoint with those of in-distribution (ID) samples, is important for reliable deep neural networks (DNNs). To achieve better detection performance, one type of approach proposes to fine-tune the model with auxiliary OOD datasets to amplify the difference between ID and OOD data through a separation loss defined on model outputs. However, none of these studies consider enlarging the feature disparity, which should be more effective compared to outputs. The main difficulty lies in the diversity of OOD samples, which makes it hard to describe their feature distribution, let alone design losses to separate them from ID features. In this paper, we neatly fence off the problem based on an aggregation property of ID features named Neural Collapse (NC). NC means that the penultimate features of ID samples within a class are nearly identical to the last layer weight of the corresponding class. Based on this property, we propose a simple but effective loss called Separation Loss, which binds the features of OOD data in a subspace orthogonal to the principal subspace of ID features formed by NC. In this way, the features of ID and OOD samples are separated by different dimensions. By optimizing the feature separation loss rather than purely enlarging output differences, our detection achieves SOTA performance on CIFAR10, CIFAR100 and ImageNet benchmarks without any additional data augmentation or sampling, demonstrating the importance of feature separation in OOD detection. Code is available at https://github.com/Wuyingwen/Pursuing-Feature-Separation-for-OOD-Detection.

## 1 Introduction

In the open world, deep neural networks (DNNs) encounter a diverse range of input images, including in-distribution (ID) data that shares the same distribution as the training data, and out-of-distribution (OOD) data, which has labels that are disjoint from those of the ID cases. Facing the complex input environment, a reliable network system must not only provide accurate predictions for ID data but also recognize unseen OOD data. This necessity gives rise to the critical problem of OOD detection (Cao et al., 2007; Liu et al., 2021), which has garnered significant attention in recent years, particularly in safety-critical applications.

A rich line of studies detect OOD samples by exploring the differences between ID and OOD data in terms of model outputs (Hendrycks & Gimpel, 2016; Liu et al., 2020), features (Sun et al., 2021; Zhu et al., 2022b; Sun et al., 2022b), or gradients (Huang et al., 2021; Wu et al., 2023). However, it has been observed that models trained solely on ID data can make over-confident predictions on OOD data, and the features of OOD data intermingle with those of ID features (Hendrycks & Gimpel, 2016; Sun et al., 2022b). To develop more effective detection algorithms, a category of works focus on the utilization of auxiliary OOD datasets, which significantly improves detection performance on unseen OOD data. One classical method, called Outlier Exposure (OE, Hendrycks et al. (2018)), employs a cross-entropy loss between the outputs of OOD data and uniformly distributed labels to fine-tune the model. Additionally, Energy method (Liu et al., 2020) proposes using the energy function as its training loss and designs an energy gap between ID and OOD data. Building on these proposed losses, recent works have concentrated on improving the quality of auxiliary OOD datasets through

---
[*]Corresponding author

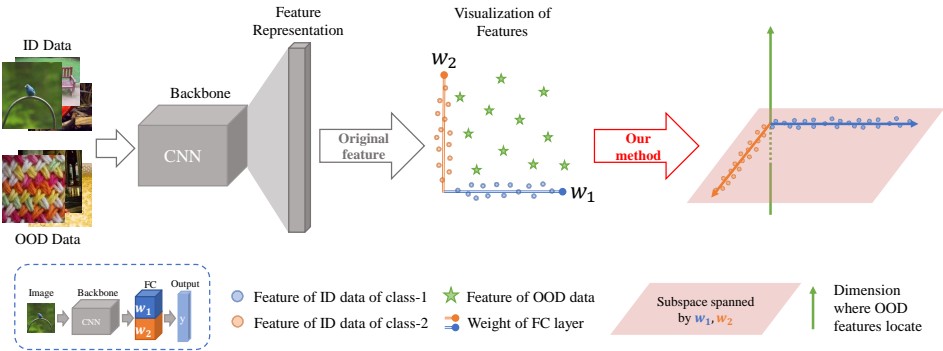

Figure 1: Overview of our method. An example of a well-trained binary classification network, where $w_i$ denotes the $i$-th weight of the last fully connected layer. The features of ID samples within a class are nearly identical to the weight of the corresponding class, which is called Neural Collapse. Based on this property, we propose to constrain OOD features on dimensions orthogonal to FC weight subspace to explicitly separate the feature manifolds between ID and OOD data.

data augmentation (Wang et al., 2024; 2023; Zheng et al., 2024) or data sampling (Ming et al., 2022a; Chen et al., 2021; Jiang et al., 2023) to achieve better detection performance.

Existing losses designed for auxiliary OOD data primarily focus on increasing the output discrepancy between ID and OOD samples (Hendrycks et al., 2018; Liu et al., 2020). However, **NONE** of these approaches consider enhancing the separability in the feature space. Insights from knowledge distillation (Gou et al., 2021) and contrastive learning (Le-Khac et al., 2020) have demonstrated that optimizing compactness or dispersion in the feature space is equally or even more important than enforcing similar constraints in the output space. Furthermore, previous detection score function design has shown the importance of employing feature information (Sun et al., 2021; 2022b; Zhu et al., 2022b), which can greatly improve detection performance. Therefore, when tackling the fine-tuning problem using auxiliary OOD data, we propose that it is crucial to **separate the features between ID and OOD data**, rather than merely enlarging their output differences.

Designing an effective feature separation loss for ID and OOD data is inherently challenging due to the diversity of OOD samples that belong to various categories. This diversity results in a dispersion of their features and difficulty in describing their feature distribution. Consequently, common feature separation losses, such as maximizing the distance between the average features of different classes (Ming et al., 2022b) or increasing the Kullback-Leibler divergence between ID and OOD distributions (Kullback, 1997), are not suitable in our cases. Despite the intricate distribution of OOD features posing a significant obstacle, in this paper, we derive solutions from the properties of ID features.

A recent observation named Neural Collapse (Papyan et al., 2020) gives us an inspiration, which reveals that the penultimate features of ID samples within a class are nearly identical to the last fully connected (FC) layer's weights of the corresponding class. Conversely, the features of OOD samples are scattered haphazardly throughout the feature space. A direct illustration [1] can be seen in Figure 1. Leveraging the property of ID features, we propose to constrain the features of OOD data on dimensions orthogonal to the subspace (denoted as $W$) spanned by FC weights. The dimension of $W$ equals to the number of ID categories, while the overall feature space dimension is significantly larger. Consequently, there are numerous redundant dimensions available for OOD features, indicating the feasibility of our method. To pursue this orthogonality, we introduce a loss function named Separation Loss (*ref.* Eq. 3), which calculates the absolute value of cosine similarity between OOD features and the weights of the final FC layer. By optimizing this simple yet effective loss to zero, we ensure that OOD features are distributed in entirely different dimensions from ID features, thereby enhancing their separability. Our approach utilizes the NC property of ID features, allowing us to avoid modeling OOD feature distributions while effectively segregating ID and OOD features. The overall method can be widely applied as a stronger baseline compared to OE (Hendrycks et al., 2018), and seamlessly integrated with other approaches like ATOM(Chen et al., 2021), POEM(Ming et al., 2022a), etc(Wang et al., 2024; 2023) by replacing the OE loss with our separation loss.

---

[1]For binary classification, NC indicates that the angle between $w_1$ and $w_2$ should be $180°$. But in order to show the general case of higher dimensions, we depict an angle of $90°$ in the figure.

We conduct extensive experiments over representative OOD detection setups, achieving the SOTA performance without any data augmentation or sampling algorithms (Ming et al., 2022a; Wang et al., 2023) on CIFAR10 (Krizhevsky et al., 2009b), CIFAR100 (Krizhevsky et al., 2009b), and ImageNet (Deng et al., 2009a) benchmarks. For example, on the CIFAR100 benchmark, by using our feature separation loss, we achieve the average FPR95 of $29.58\%$ and AUROC of $94.01\%$, outperforming the traditional OE (Hendrycks et al., 2018) method by $8.19\%$ on FPR95. The contribution of our paper is summarized as follows:

- We are the first to propose the concept of feature separation when using auxiliary OOD data to fine-tune models, while previous works pay more attention to the output separation, providing new insights into the design of OOD data loss functions.

- To overcome the difficulty caused by OOD data diversity, we propose a feature separation loss based on the neural collapse property of ID features, which constrains OOD features to lie in dimensions where ID features are scarcely distributed.

- Our SOTA detection performance on representative OOD detection settings verify the effectiveness of our feature separation loss, implying that our loss can be a stronger baseline for future researches.

## 2 RELATED WORK

**Post-hoc Detection.** Given a model that is only trained by ID data, post-hoc detection approaches design score functions based on it to distinguish ID and OOD data. One type method named density-based (Lee et al., 2018; Kobyzev et al., 2020; Zisselman & Tamar, 2020; Kingma & Dhariwal, 2018; Jiang et al., 2021; Choi et al., 2018) is to explicitly model the ID data with some probabilistic models and flag test data in low-density regions as OOD samples. More popular approaches are to derive confidence score based on model outputs (Hendrycks & Gimpel, 2016; Liang et al., 2017; Liu et al., 2020), features (Sun et al., 2021; Zhu et al., 2022b; Sun et al., 2022b; Lee et al., 2018; Ndiour et al., 2020; Cook et al., 2020; Ndiour et al., 2020; Cook et al., 2020; Wang et al., 2022) or gradients (Huang et al., 2021; Wu et al., 2023; Lee et al., 2023; Lust & Condurache, 2020; Sun et al., 2022a; Igoe et al., 2022). For example, the recent feature-distance based method KNN Sun et al. (2022b) employs the Euclidean distance to the $k$-th nearest neighborhood of training data as a measurement to detect OOD data. Different from KNN which designs a detection metric based on fixed feature representations, our approach explicitly enlarges the feature distance between ID and OOD data through optimization.

**Contrastive Learning based Detection.** Different from post-hoc methods based on vanilla-trained models, such methods generally apply contrastive losses defined on ID data in the model training process to obtain better feature representations for OOD detection. For example, KNN+ (Sun et al., 2022b) utilizes the SupCon loss (Khosla et al., 2020), which encourages alignment of features within a class and dispersion of features of different classes, to train a network to obtain greater differentiation between ID and OOD samples. Besides, CSI (Tack et al., 2020) contrasts original samples with their distributionally-shifted augmentations to improve detection performance. Recent advancements, such as CIDER (Ming et al., 2022b), combine a compactness loss to cluster samples near their class prototypes and a dispersion loss to maximize angular distances between different class prototypes, providing a more direct and clearer geometric interpretation for the disparity between ID and OOD samples.

**Auxiliary OOD Data based Detection.** With access to part of OOD data, previous works design training algorithms to utilize auxiliary OOD data for OOD detection. One type method is to propose unsupervised training loss functions (Hendrycks et al., 2018; Liu et al., 2020; Bai et al., 2023), such as the Kullback-Leibler divergence between OOD output probability and uniformly distributed label (Hendrycks et al., 2018), to fine-tune the model. Based on the proposed losses, another type is to select OOD data close to the decision boundary (Ming et al., 2022a) or conduct data augmentation through adversarial attack (Chen et al., 2021; Wang et al., 2024) and model perturbations (Wang et al., 2023) in the training process, which can tight the boundary so that pushing unseen OOD data far away from it. In general, using auxiliary OOD data in the training process can significantly improve detection performance, achieving better results compared with other detection approaches.

## 3 METHOD

### 3.1 PRELIMINARY

**OOD Detection Problem.** The framework for OOD detection is outlined as follows. We consider a classification problem involving $C$ classes, where $\mathcal{X}$ represents the input space and $\mathcal{Y}$ denotes the label space. The joint data distribution over $\mathcal{X} \times \mathcal{Y}$ is referred to as $D_{\mathcal{X}\mathcal{Y}}$. Let $f_\theta : \mathcal{X} \mapsto \mathcal{Y}$ be a model trained on samples drawn independently and identically distributed (*i.i.d.*) from $D_{\mathcal{X}\mathcal{Y}}$ with parameters $\theta$. Then, the distribution of ID data is the marginal distribution of $D_{\mathcal{X}\mathcal{Y}}$ over $\mathcal{X}$, denoted as $D_{in}$. Conversely, the distribution of OOD data is represented as $D_{out}$, whose label set does not intersect with $\mathcal{Y}$. The primary objective of OOD detection is to determine whether a test input $x$ originates from $D_{in}$ or $D_{out}$. Typically, this decision is made using a score function $S$ as follows:

$$G_\lambda(x) = \begin{cases} \text{ID} & \text{if } S(x, f) \geq \lambda \\ \text{OOD} & \text{if } S(x, f) \leq \lambda \end{cases}$$

where $\lambda$ is a threshold. Samples with scores higher than $\lambda$ are classified as ID data. The threshold is usually set based on ID data to ensure that a high fraction of ID data (e.g., 95%) is correctly identified as ID samples.

**Finetune Model with Auxiliary OOD Data.** In this paper, we consider the task of using auxiliary OOD data to fine-tune the model (Hendrycks et al., 2018; Liu et al., 2020; Ming et al., 2022a; Wang et al., 2024), which can effectively enlarge the discrepancy between ID and unseen OOD data. Let's denote the auxiliary OOD dataset as $D_{\text{out}}^{\text{aux}}$, which is a subset of real OOD datasets but has different distributions from the test OOD datasets in the experiments for fair comparison. One classical method is the Outlier Exposure (OE, (Hendrycks et al., 2018)), which designs an outlier exposure loss that calculates the cross-entropy function between OOD outputs and uniformly distributed labels. The equation is as follows:

$$L_{\text{OE}}(x) = -\frac{1}{C} \sum_{j=1}^{C} \log f_j(x), \tag{1}$$

where $f_j(x)$ denotes the $j$-th element of the model output $f(x)$. The final training objective of OE is to simultaneously minimize cross-entropy loss on ID data and outlier exposure loss on OOD data, which can be formalized as:

$$\min_f \mathbb{E}_{(x,y)\sim D_{\text{in}}} L_{\text{CE}}(x, y) + \lambda \mathbb{E}_{x\sim D_{\text{out}}^{\text{aux}}} L_{\text{OE}}(x) \tag{2}$$

where $\lambda$ is a hyper-parameter. This optimization problem is regarded as a basic setting in auxiliary OOD data approaches. Most of subsequent methods adopt the same or similar loss functions that encourage ID and OOD data to differ in the output space. For example, POEM (Ming et al., 2022a) designs a data sampling algorithm for efficient training, and DAL (Wang et al., 2024) employs adversarial features to calculate the OE loss to minimize the generalization gap between auxiliary and real unseen OOD data.

### 3.2 MOTIVATION

Previous works have focused on increasing the discrepancy between ID and OOD data in the output space, while in this paper, we propose to explicitly enlarge the disparity of their features. Intuitively, separating features of ID and OOD data should be beneficial to OOD detection compared to solely augmenting the output differences. Existing feature separation functions in other fields, such as the dispersion loss that enlarges the distance between the average features of different classes (Ming et al., 2022b; Khosla et al., 2020), are not suitable for diverse OOD data since their features are dispersed instead of clustering around the mean. To design a separation loss that can handle the complicated distribution of OOD features, we delve into the property of ID features. A recent observation named NC (Papyan et al., 2020) gives us a new insight, which reveals that the penultimate features of ID samples within a class are nearly identical to the last layer weight of the corresponding class. This intriguing property has stimulated many fields of research, including low-dimensional characteristics of ID features (Garrod & Keating, 2024; Rangamani et al., 2023) and model generalization analysis (Kothapalli, 2022; Hui et al., 2022). Particularly, several works employ the principal component spaces identified by NC to design detection score functions (Liu & Qin, 2023; Zhang et al., 2024b;

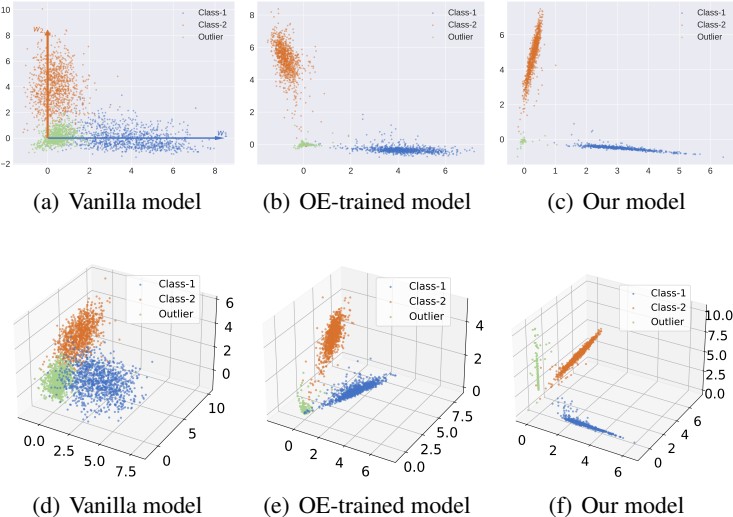

Figure 2: Visualization of features projected into the two-dimensional space consisted of $w_1$ and $w_2$ (*ref. Figure 1*) and the three-dimensional space consisted of $w_1$, $w_2$ and the principal eigenvector of OOD features on CIFAR10 benchmark. The Class-1 and Class-2 represent features of test ID samples of class-1 and class-2, and the Outlier means features of test unseen OOD data, *i.e.* SVHN. It can be observed that the feature separability between ID and OOD data gradually increases from left (Vanilla model) to right (Our model).

Haas et al., 2022; Ammar et al., 2023), which demonstrates the large potential of NC applied in OOD detection. We conduct empirical experiments on CIFAR10 to validate the NC property, as Figure 2(a) shows, where we plot the feature distribution of test ID samples under weight basis space (For an accurate visualization, we select two class samples in CIFAR10 since the principal component dimension of their features equals to the number of their classes, *i.e.* two, according to NC property. More visualization results of samples of other classes can be seen in Appendix A.9). Furthermore, with the comparison of OOD features of vanilla and OE-trained model (Figure 2(a) *vs.* Figure 2(b)), we discover that although the outlier exposure loss only optimizes the output of training OOD samples, it implicitly changes the distribution of test unseen OOD features, making them more clustered and far way from test ID features. However, from the 3D visualization of features in Figure 2(e), it can be observed that the features of test unseen OOD data almost lie in the same subspace as ID features, without taking advantage of the new dimension (z-axis) to further widen the ID-OOD difference. Based on the above observations, we then design a feature separation loss without modeling OOD feature distributions as follows.

### 3.3 FEATURE SEPARATION LOSS

Our key idea is to confine the features of OOD data to dimensions where ID features are sparsely distributed. Considering that the principal subspace of ID features is $C$-dimensional, as determined by the NC property, while the overall feature space has a significantly larger dimensionality, there exist ample redundant dimensions that can accommodate OOD features. In pursuit of our goal, a straightforward condition arises: $z^T w_i = 0, i = 1, 2, ..., C$, where $z$ denotes the normalized feature of OOD data, and $w_i$ denotes the normalized fully connected layer weight for class $i$. According to this condition, we devise a Separation Loss for OOD data, which computes the average absolute value of the cosine similarity between $z$ and $w_i$. The specific equation is as follows:

$$L_{\text{Sep}} = \frac{1}{C} \sum_{i=1}^{C} \left| z^T w_i \right| \tag{3}$$

Through minimizing the $L_{\text{Sep}}$ loss, the OOD feature $z$ tends to be distributed in the dimensions that are orthogonal to $w_i, i = 1, 2, ..., C$. Figures 2(c) and 2(f) illustrate the features of our model fine-tuned using $L_{\text{Sep}}$. As observed, the features of outlier samples are indeed distributed in different dimensions from $w_i$, resulting in a larger discrepancy between ID and OOD features. Except for

the $L_{\text{Sep}}$ loss, we also propose an assistant loss that encourages ID features (denote as $z_{\text{ID}}$) within a class to align closely with the FC weight of their corresponding class (denote as $w_y$). We term this loss function as $L_{\text{Clu}}$ as it promotes neural collapse phenomenon (Papyan et al., 2020), enabling ID features within a class more clustered. The formulation is as follows:

$$L_{\text{Clu}} = -z_{ID}^T w_y \tag{4}$$

Our empirical experiment in Sec 4.4 indicates that adding $L_{\text{Clu}}$ in the training loss can further improve the detection performance. Combining the above two losses, the final optimization problem can be formulated as:

$$\min_f \mathbb{E}_{(x,y) \sim D_{in}}(L_{\text{CE}} + \alpha L_{\text{Clu}}) + \mathbb{E}_{x \sim D_{out}^{aux}}(\lambda L_{\text{OE}} + \beta L_{\text{Sep}}) \tag{5}$$

where $\alpha$, $\lambda$ and $\beta$ are hyper-parameters. In our experiments, we use the common setting $\lambda = 0.5$ in previous works (Hendrycks et al., 2018) and set $\alpha = 1.0$ and $\beta = 1.0$ for simplicity.

### 3.4 OVERALL FRAMEWORK

**Train.** Based on the optimization problem outlined in Eq. 5, we detail the training procedure as follows. Our training process is consisted of two stage: Stage I-training with cross-entropy loss to ensure the occurrence of NC phenomenon; Stage II-training with all losses including cross-entropy, outlier exposure, and our proposed separation and clustering losses to enlarge the discrepancy between ID and OOD data. Notably, when a well-trained model is used as the initial parameter, Stage I is not necessary and thus can be omitted. More detailed strategy choices and their impact on our performance can be seen in Appendix A.5.

**Test.** After fine-tuning the model with our loss, we propose a new score function to detect OOD samples. Since our method simultaneously optimizes the outputs and features of OOD data, a more proper score function is the sum of the traditional MSP (Hendrycks & Gimpel, 2016) and the average cosine similarity between features and $w_i$. Mathematically, it can be expressed as:

$$S(x,f) = \max_i \frac{e^{y_i}}{\sum_j^C e^{y_j}} + \frac{1}{C} \sum_{i=1}^C |z^T w_i| \tag{6}$$

where $y$ denotes the model output $f(x)$. Our experiments in Appendix A.8 compare the performance of using our score function with only using the MSP score. The result indicates that our method also performs well under MSP score, but slightly better under our proposed score function.

## 4 EXPERIMENTS

In this section, we first conduct experiments on CIFAR10, CIFAR100 and ImageNet benchmarks to validate the superiority of our method in Sec 4.1. Then, we consider a variety of model architectures to further verify the effectiveness of our method in Sec 4.2. Subsequently, we study the hyper-parameter sensitivity of our method in Sec 4.3 and explore the contribution of each loss part in Sec 4.4. In the last part, we discuss the loss design based on different distance metrics, numerical degree of feature separation and requirement on feature dimension in Sec 4.5. In our Appendix, we report more experimental results, including combining our loss with Energy-OE (Liu et al., 2020) in Appendix A.1, detailed results on different networks in Appendix A.2, performance on hard OOD detection settings proposed in CSI (Tack et al., 2020) in Appendix A.3, generalization to class-imbalanced datasets in Appendix A.4, training strategy choice and impact in Appendix A.5, feature distance anchoring on class mean in Appendix A.6, fluctuation of performance in Appendix A.7, influence of different score functions in Appendix A.8, and more visualization results in Appendix A.9. To begin with, we introduce our experiment setups as follows.

**OOD Datasets.** For CIFAR benchmarks, we randomly choose 300K samples from the 80 Million Tiny Images (Torralba et al., 2008) as our auxiliary OOD dataset. And we adopt five routinely used datasets as the test OOD datasets, including SVHN (Netzer et al., 2011), LSUN (Yu et al., 2015), iSUN (Xu et al., 2015), Texture (Cimpoi et al., 2014) and Places365 (Zhou et al., 2017), which have non-overlapping categories *w.r.t.* CIFAR datasets. For ImageNet benchmark, we use a validation subset of ImageNet-21k-p dataset as auxiliary OOD dataset. And we adopt four commonly-used

OOD datasets for evaluation, including iNaturalist (Van Horn et al., 2018), SUN (Xiao et al., 2010), Places (Zhou et al., 2017) and Textures (Cimpoi et al., 2014).

**Pre-training Setups.** For CIFAR benchmarks, we employ Wide ResNet-40-2 (Zagoruyko & Komodakis, 2016) trained for 200 epochs, with batch size 128, init learning rate 0.1, momentum 0.9, weight decay 0.0005, and cosine schedule. For ImageNet benchmarks, we directly use the pre-trained ResNet50 (He et al., 2016) model in Pytorch as the baseline network.

**Fine-tuning Setups.** For both CIFAR10 and CIFAR100 benchmarks, we adopt the model parameter of the 99th epoch in the pre-training process as our initial network parameters, and then add auxiliary OOD data to train the model for 50 epochs with ID batch size 128, OOD batch size 256, initial learning rate 0.07, momentum 0.9, weight decay 0.0005 and cosine schedule. This setting is aligned with experiments in DAL (Wang et al., 2024). Since the initial model is not sufficiently converged, we add our proposed $L_{\text{Sep}}$ and $L_{\text{Clu}}$ into the training loss after 25th epoch of the whole fine-tuning stage. For ImageNet benchmark, we use the pre-trained model in Pytorch as initial network, and then fine-tune the model for 5 epochs with ID/OOD batch size 64, initial learning rate $1e-4$, momentum 0.9, weight decay 0.0005 and cosine schedule.

**Compared Methods.** We compare our method with post-hoc approaches, contrastive learning based methods, and auxiliary OOD data based methods. The post-hoc methods include MSP (Hendrycks & Gimpel, 2016), Energy (Liu et al., 2020), Maha (Lee et al., 2018), and KNN (Sun et al., 2022b). The contrastive learning based methods include CSI (Tack et al., 2020), CIDER (Ming et al., 2022b), and KNN+ (Sun et al., 2022b). The auxiliary OOD data based methods include OE (Hendrycks et al., 2018), Energy-OE (Liu et al., 2020), POEM (Ming et al., 2022a), and DAL (Wang et al., 2024). For OE and Energy-OE, we adopt the same training setting as ours, since we have discovered that their recommended setting in the original paper performs much worse than our setting. For other methods, we adopt their suggested setups but unify the backbones for fairness.

**Evaluation Metrics.** We report two classical metrics in this paper: 1) FPR95: the false positive rate of OOD samples when the true positive rate of ID samples is at 95%. 2) AUROC: the area under the receiver operating characteristic curve. A lower FPR95 and a higher AUROC indicate better detection performance.

## 4.1 MAIN RESULTS

The main results are shown in Table 1 and Table 2, where we report the FPR95 and AUROC across the considered real OOD datasets [2]. Compared to methods based on vanilla or contrastive learning models, whose training datasets only contain ID samples, incorporating auxiliary OOD data into the training process can significantly reduce the FPR95 and improve the AUROC, indicating that this direction is valuable to explore. Compared to the classical OE approach (Hendrycks et al., 2018), our method reduces the average FPR95 by 0.87% on CIFAR10, 8.30% on CIFAR100, and 2.93% on ImageNet, just by adding our Separation and Cluster losses into the training procedure. This result demonstrates the effectiveness of our proposed losses. In addition to the OE approach, we also compare our method with other advanced works, including the classical work that studies data sampling strategies (POEM, (Ming et al., 2022a)) and the adversarial feature augmentation work that aims to mitigate the impact of OOD distribution discrepancy (DAL, (Wang et al., 2024)). It is worth noticing that our method does not employ any data augmentation or selection algorithms, while exhibiting superior performances on CIFAR and ImageNet benchmarks. On CIFAR100, our method outperforms the best baseline DAL by 2.00% on FPR95 and 1.18% on AUROC. Notably, the performance of DAL method on the Places dataset of ImageNet benchmark is vastly different from their reported result in the original DAL paper. The reason stems from our use of a curated subset (Huang & Li, 2021) from Places365, which contains some Near-OOD classes semantically similar to ImageNet categories (e.g., hayfield vs. hay, cornfield vs. corn). In contrast, the DAL paper likely evaluated on randomly sampled Far-OOD data from the complete 10 million size Places365 dataset. Based on the outstanding performance of our method, we suggest that the feature separation loss, which is simple yet effective, can be used as a basic training function like OE loss (Hendrycks et al., 2018) in the further works.

---

[2] The symbol $^*$ in the table means the results are cited from DAL (Wang et al., 2024)

Table 1: Results on ImageNet-1k benchmark with auxiliary OOD data. The best result is in bold.

| Model | Method | Far-OOD Datasets | | | | Near-OOD Datasets | | | | Average | | ID Acc↑ |
| | | iNaturalist | | Textures | | SUN | | Places | | | | |
| | | FPR95↓ | AUROC↑ | FPR95↓ | AUROC↑ | FPR95↓ | AUROC↑ | FPR95↓ | AUROC↑ | FPR95↓ | AUROC↑ | |
| ResNet50 | OEHendrycks et al. (2018) | 48.30 | 88.91 | 58.60 | 82.78 | 61.40 | 83.09 | 70.36 | 80.78 | 59.66 | 83.89 | 76.04 |
| | DALWang et al. (2024) | 47.92 | 89.12 | 57.91 | 83.02 | 61.20 | 83.22 | 70.55 | 80.79 | 59.39 | 84.04 | 75.94 |
| | Ours | 43.01 | 90.17 | 55.35 | 83.45 | 60.11 | 83.56 | 68.46 | 81.31 | **56.73** | **84.62** | **76.10** |
| ViT-B-16 | OEHendrycks et al. (2018) | 41.96 | 90.49 | 52.25 | 85.97 | 65.61 | 82.30 | 70.20 | 80.93 | 57.51 | 84.92 | 80.05 |
| | DALWang et al. (2024) | 40.52 | 90.92 | 50.94 | 86.20 | 65.07 | 82.39 | 70.17 | 80.96 | **56.67** | 85.12 | 80.06 |
| | Ours | 40.10 | 91.06 | 51.70 | 86.13 | 65.58 | 82.25 | 70.12 | 81.07 | 56.88 | **85.13** | **80.29** |

Table 2: Results on CIFAR10 and CIFAR100 benchmarks with WideResNet-40-2 model. The best result is in bold.

| Method | Far-OOD Datasets | | | | | | | | | | Average | |
| | SVHN | | LSUN | | iSUN | | Textures | | Places365 | | | |
| | FPR95↓ | AUROC↑ | FPR95↓ | AUROC↑ | FPR95↓ | AUROC↑ | FPR95↓ | AUROC↑ | FPR95↓ | AUROC↑ | FPR95↓ | AUROC↑ |
| CIFAR-10 | | | | | | | | | | | | |
| With vanilla training | | | | | | | | | | | | |
| MSPHendrycks & Gimpel (2016) | 44.22 | 93.61 | 27.56 | 96.12 | 69.62 | 85.29 | 60.02 | 88.53 | 65.68 | 86.25 | 53.42 | 89.96 |
| EnergyLiu et al. (2020) | 31.81 | 94.65 | 4.6 | 98.96 | 50.06 | 89.75 | 49.68 | 90.09 | 42.28 | 90.82 | 35.69 | 92.85 |
| MahaLee et al. (2018) | 42.67 | 90.71 | 18.96 | 96.46 | 28.86 | 93.76 | 26.22 | 92.81 | 86.78 | 69.14 | 40.70 | 88.58 |
| KNNSun et al. (2022b) | 44.76 | 92.55 | 27.38 | 95.34 | 43.84 | 91.24 | 37.64 | 92.82 | 49.23 | 87.89 | 40.57 | 91.97 |
| With contrastive learning | | | | | | | | | | | | |
| CSI*Tack et al. (2020) | 17.37 | 97.69 | 6.75 | 98.46 | 12.58 | 97.95 | 25.65 | 94.70 | 40.00 | 92.05 | 20.47 | 96.17 |
| CIDERMing et al. (2022b) | 6.76 | 98.44 | 7.45 | 98.76 | 26.03 | 95.93 | 22.85 | 95.75 | 43.70 | 91.94 | 21.36 | 96.16 |
| KNN+*Sun et al. (2022b) | 3.28 | 99.33 | 2.24 | 98.90 | 17.85 | 97.65 | 10.87 | 97.92 | 30.63 | 94.98 | 12.97 | 97.32 |
| With auxiliary OOD data | | | | | | | | | | | | |
| OEHendrycks et al. (2018) | 1.40 | 99.54 | 0.85 | 99.64 | 2.20 | 99.26 | 2.80 | 99.26 | 9.55 | 97.39 | 3.36 | **99.02** |
| Energy-OELiu et al. (2020) | 0.75 | 99.50 | 0.90 | 98.98 | 1.50 | 99.22 | 2.75 | 98.92 | 9.05 | 97.33 | 2.99 | 98.79 |
| POEMMing et al. (2022a) | 25.66 | 95.43 | 94.97 | 76.44 | 1.58 | 99.64 | 20.62 | 95.73 | 53.39 | 88.38 | 39.24 | 91.10 |
| DALWang et al. (2024) | 0.75 | 99.28 | 0.75 | 99.62 | 0.70 | 99.33 | 2.35 | 98.99 | 8.90 | 97.10 | 2.69 | 98.86 |
| Ours | 0.40 | 99.28 | 0.60 | 99.68 | 1.60 | 99.25 | 2.45 | 98.83 | 7.40 | 97.60 | **2.49** | 98.93 |
| CIFAR-100 | | | | | | | | | | | | |
| With vanilla training | | | | | | | | | | | | |
| MSPHendrycks & Gimpel (2016) | 74.79 | 79.64 | 54.72 | 86.46 | 93.85 | 56.92 | 88.76 | 68.48 | 83.24 | 71.95 | 79.07 | 72.69 |
| EnergyLiu et al. (2020) | 70.18 | 87.15 | 17.15 | 97.05 | 91.37 | 65.50 | 84.77 | 76.72 | 78.91 | 75.77 | 62.75 | 80.44 |
| MahaLee et al. (2018) | 77.73 | 78.01 | 98.46 | 63.44 | 47.74 | 88.76 | 54.93 | 82.53 | 97.22 | 54.11 | 75.22 | 73.37 |
| KNNSun et al. (2022b) | 71.86 | 83.31 | 78.89 | 70.09 | 79.60 | 70.86 | 72.89 | 80.05 | 80.91 | 71.33 | 76.83 | 75.13 |
| With contrastive learning | | | | | | | | | | | | |
| CSI*Tack et al. (2020) | 64.50 | 84.62 | 25.88 | 95.93 | 70.62 | 80.83 | 61.50 | 86.74 | 83.08 | 77.11 | 61.12 | 95.05 |
| CIDERMing et al. (2022b) | 16.47 | 96.23 | 45.45 | 81.64 | 66.01 | 82.21 | 49.79 | 87.48 | 82.66 | 68.39 | 52.08 | 83.19 |
| KNN+*Sun et al. (2022b) | 32.50 | 93.86 | 47.41 | 84.93 | 39.82 | 91.12 | 43.05 | 88.55 | 63.26 | 79.28 | 45.20 | 87.55 |
| With auxiliary OOD data | | | | | | | | | | | | |
| OEHendrycks et al. (2018) | 38.70 | 92.90 | 18.30 | 96.67 | 36.35 | 92.59 | 43.05 | 91.00 | 52.45 | 87.86 | 37.77 | 92.21 |
| Energy-OELiu et al. (2020) | 17.75 | 96.94 | 34.00 | 94.82 | 60.75 | 87.32 | 45.70 | 90.09 | 53.50 | 89.08 | 42.34 | 91.65 |
| POEMMing et al. (2022a) | 45.41 | 90.70 | 3.01 | 99.24 | 18.60 | 95.79 | 51.37 | 83.85 | 84.13 | 73.93 | 40.5 | 88.87 |
| DALWang et al. (2024) | 16.45 | 96.10 | 17.00 | 96.52 | 36.95 | 90.88 | 38.40 | 91.72 | 48.55 | 88.91 | 31.47 | 92.82 |
| Ours | 17.95 | 96.52 | 12.50 | 97.64 | 27.00 | 93.85 | 41.70 | 91.37 | 48.20 | 90.64 | **29.47** | **94.00** |

## 4.2 DIFFERENT ARCHITECTURES

To further verify the effectiveness of our method, we evaluate and compare our performance with other approaches on more network architectures, including ResNet18 (He et al., 2016) and DenseNet121 (Huang et al., 2017). The results are shown in Table 3, where our method exhibits consistently superior performance across various architectures on CIFAR10 and CIFAR100 benchmarks. For instance, we reduce the FPR95 by $4.55\%$ compared to DAL (Wang et al., 2024) with ResNet18 architecture on CIFAR100 benchmark. Detailed results can be seen in Appendix A.2.

Table 3: Results on different network architectures on CIFAR10 and CIFAR100 benchmarks. We report the average FPR95/AUROC across five OOD datasets. The best result is in bold.

| Method | CIFAR-10 | | | CIFAR-100 | | |
| | WRN-40-2 | ResNet18 | DenseNet-121 | WRN-40-2 | ResNet18 | DenseNet121 |
| OEHendrycks et al. (2018) | 3.36/99.02 | 6.35/97.35 | 10.79/97.54 | 37.77/92.21 | 56.96/90.19 | 62.08/86.76 |
| DALWang et al. (2024) | 2.69/98.86 | 3.61/98.20 | 9.75/97.71 | 31.47/92.82 | 54.89/**90.95** | 61.25/87.66 |
| Ours | **2.49/98.93** | **3.52/98.75** | **8.90/97.74** | **29.47/94.00** | **50.34**/90.90 | **59.13/88.45** |

## 4.3 HYPER-PARAMETER SENSITIVITY

In this section, we study the influence of coefficient $\alpha$ and $\beta$ in Eq. 5 on the detection performance. Specifically, we evaluate our method on CIFAR10 benchmark with $\alpha \in \{0.1, 0.5, 1.0, 2.0\}$ and $\beta \in \{0.1, 0.5, 1.0, 2.0\}$. Experiment results are shown in Table 4. Notably, our approach is not sensitive to the choice of hyper-parameters. Furthermore, we discover that using $\alpha = 0.1$ and $\beta = 0.1$ can achieve better performance of our method than our previous report.

Table 4: Influence of loss coefficient $\alpha$ and $\beta$. We report the average FPR95/AUROC across five OOD datasets on CIFAR10 benchmark. The best result is in bold, and the result for the parameter used in our main experiment is underlined.

| CIFAR10 | $\beta = 0.1$ | $\beta = 0.5$ | $\beta = 1.0$ | $\beta = 2.0$ |
|---|---|---|---|---|
| $\alpha = 0.1$ | **2.21/99.13** | 2.41/99.03 | 2.40/99.12 | 2.60/98.97 |
| $\alpha = 0.5$ | 2.45/98.91 | 2.52/98.90 | 2.51/99.07 | 2.43/99.08 |
| $\alpha = 1.0$ | 2.49/98.77 | 2.34/98.92 | 2.49/98.93 | 2.44/99.01 |
| $\alpha = 2.0$ | 2.46/98.73 | 2.51/98.24 | 2.36/98.38 | 2.67/98.72 |

## 4.4 ABLATION STUDY

Considering our training objective loss contains four parts: $L_{CE}$, $L_{Clu}$, $L_{OE}$, and $L_{Sep}$, we explore the contribution of each part to the final detection performance in this section. The cross-entropy loss $L_{CE}$ is used for ensuring ID accuracy, thus we skip it when discussing the detection performance. The rest three parts, one ($L_{OE}$) is for output discrepancy and the other two ($L_{Clu}$ and $L_{Sep}$) is for feature separation. We firstly evaluate the performance of purely using cross-entropy loss ($L_{CE}$) and outlier exposure loss ($L_{OE}$), namely OE method (Hendrycks et al., 2018). And then we discuss three situations: 1) adding $L_{Clu}$; 2) adding $L_{Sep}$; 3) adding $L_{Clu}$ and $L_{Sep}$. The results are shown in Table 5. Comparing the No.1 and No.3, it shows that our feature separation loss significantly improves detection performance compared to OE method. Additionally, only using cluster loss damages the performance but integrating it with separation loss can achieve the best result. The underlying reason is that the cluster loss only controls the property of ID features, but has negligible effect on enlarging the discrepancy between ID and OOD features when purely using it.

Table 5: Performance of our method under different training losses. The cross-entropy loss is used by default in all cases.

| No. | Training Loss | CIFAR10 | | CIFAR100 | |
|---|---|---|---|---|---|
| | | FPR95↓ | AUROC↑ | FPR95↓ | AUROC↑ |
| 1 | $L_{OE}$ | 3.36 | **99.02** | 37.77 | 92.21 |
| 2 | $L_{OE}+L_{Clu}$ | 3.62 | 98.96 | 39.91 | 91.22 |
| 3 | $L_{OE}+L_{Sep}$ | 2.65 | 99.00 | 33.30 | 93.42 |
| 4 | $L_{OE}+L_{Clu}+L_{Sep}$ | **2.49** | 98.93 | **29.47** | **94.00** |

## 4.5 DISCUSSION

**Cosine Similarity *vs.* Euclidean Distance.** Our designed loss is calculated based on cosine similarity, which then induces dimensionality separation between ID and OOD features. In this part, we compare with an intuitive loss design, that is, maximizing the Euclidean distance between OOD features and weights of the last FC layer, with our orthogonality-based loss $L_{Sep}$ to illustrate the importance of utilizing redundant dimensions to enlarge feature discrepancy. Since maximizing the Euclidean distance will cause its value to approach infinity, we instead use $\frac{1}{\|z - w_i\|}$ for OOD features and $\|z_{ID} - w_y\|$ for ID features as the training loss in the Euclidean distance setting, and then minimize this loss to fine-tune the model. The comparison results are presented in Table 6, where the cosine similarity loss significantly outperforms Euclidean distance loss. The underlying reason may be that our separation loss utilizes new dimensions to separate ID and OOD features. When faced with unseen OOD data, the feature variations tend to fall on the new dimension, resulting in minimal changes on the output.

Table 6: Comparison between using the Euclidean distance and cosine similarity (ours) as the training loss to separate ID-OOD features.

| Method | SVHN | | LSUN | | iSUN | | Textures | | Places365 | | Average | |
|---|---|---|---|---|---|---|---|---|---|---|---|---|
| | FPR95↓ | AUROC↑ | FPR95↓ | AUROC↑ | FPR95↓ | AUROC↑ | FPR95↓ | AUROC↑ | FPR95↓ | AUROC↑ | FPR95↓ | AUROC↑ |
| CIFAR-10 | | | | | | | | | | | | |
| Euclidean | 1.70 | 99.48 | 1.15 | 99.60 | 3.20 | 99.22 | 4.55 | 98.95 | 12.55 | 95.97 | 4.63 | 98.64 |
| Ours | 0.40 | 99.28 | 0.60 | 99.68 | 1.60 | 99.25 | 2.45 | 98.83 | 7.40 | 97.60 | **2.49** | **98.93** |
| CIFAR-100 | | | | | | | | | | | | |
| Euclidean | 51.95 | 85.97 | 19.95 | 96.13 | 42.35 | 87.45 | 44.80 | 88.18 | 56.95 | 85.29 | 43.20 | 88.60 |
| Ours | 17.95 | 96.52 | 12.50 | 97.64 | 27.00 | 93.85 | 41.70 | 91.37 | 48.20 | 90.64 | **29.47** | **94.00** |

**Feature Separation Degree.** To validate the effectiveness of our proposed loss, we evaluate the feature separation degree between ID and OOD data. Leveraging the NC property of ID features, we design metrics based on the fully connected layer weights: (1) Euclidean distance between features and the predicted class weight, (2) cosine similarity between features and the predicted class weight, and (3) reconstruction error to the subspace spanned by FC weights. As shown in Table 7, our model exhibits significantly larger differences under these metrics compared to vanilla and OE-trained models, where OOD refers to unseen test OOD data. Additional experiments using class mean vectors for distance measurement are provided in Appendix A.6.

Table 7: Feature separation degree of different methods measured by three metrics. The higher difference (Diff) means better discrepancy between ID and OOD features.

| Method | Euclidean Distance | | | Cosine Similarity | | | Reconstruction Error | | |
|---|---|---|---|---|---|---|---|---|---|
| | ID | OOD | Diff↑ | ID | OOD | Diff↑ | ID | OOD | Diff↑ |
| Vanilla | 0.80 | 1.12 | 0.32 | 0.69 | 0.47 | 0.22 | 0.19 | 0.32 | 0.13 |
| OE | 0.86 | 1.21 | 0.35 | 0.79 | 0.32 | 0.47 | 0.41 | 0.83 | 0.42 |
| Ours | 0.69 | 1.16 | **0.47** | 0.75 | $3e^{-5}$ | **0.75** | 0.43 | 0.86 | **0.43** |

**Requirement on Feature Dimension** Our method leverages redundant dimensions in the feature space to enhance the discrepancy between ID and OOD features, requiring the feature dimension to exceed the output dimension. This condition typically satisfied in modern CNNs, given the class counts of common datasets (ranging from tens to thousands). However, we also evaluate scenarios where feature dimensions are smaller than output dimensions. Experiments on CIFAR100 using WideResNet-40-1 (feature dimension = 64) demonstrate that, while less pronounced than in high-dimensional feature cases, our loss still marginally outperforms the basic OE loss (see Table 8). Future work could explore applying our loss to intermediate layers, where NC phenomenon also occurs Rangamani et al. (2023); Parker et al. (2023), or inserting additional linear layers to increase feature dimensions.

Table 8: Results on WideResNet-40-1 model (feature dimension is 64) on CIFAR100 benchmark. We report FPR95/AUROC.

| Method | SVHN | | LSUN | | iSUN | | Textures | | Places365 | | Average | |
|---|---|---|---|---|---|---|---|---|---|---|---|---|
| | FPR95↓ | AUROC↑ | FPR95↓ | AUROC↑ | FPR95↓ | AUROC↑ | FPR95↓ | AUROC↑ | FPR95↓ | AUROC↑ | FPR95↓ | AUROC↑ |
| OE Hendrycks et al. (2018) | 40.60 | 92.19 | 20.30 | 96.13 | 55.70 | 86.81 | 55.90 | 85.44 | 59.55 | 84.65 | 46.41 | 89.04 |
| Ours | 37.05 | 92.27 | 20.30 | 96.08 | 56.65 | 85.17 | 48.00 | 87.37 | 59.30 | 84.54 | **44.26** | **89.09** |

## 5 LIMITATION

The main limitation lies on two aspects: one is the requirement on feature dimension, which we have discussed; the other is our dependency on NC property. Although for modern CNNs and language models, NC is a relatively common phenomenon Zhu et al. (2024); Dang et al. (2023) that happens when networks convergence on training dataset. But there will still be cases where the NC property may fail Hui et al. (2022), like imbalanced data distribution (we discussed this case in Appendix A.4). Our method essentially utilize the low-dimensional property of ID features, which induces from Neural Collapse. Therefore, when NC fails, it is important to verify whether the low dimension of the feature exists and how to determine the low dimensional subspace, which is an important and interesting direction for our future work.

## 6 CONCLUSION

In this paper, we propose a novel training loss to enhance feature discrepancy between ID and OOD data during model fine-tuning with auxiliary OOD datasets. Leveraging the Neural Collapse property of ID features—where penultimate features of ID samples converge to their class weights—we introduce a separation loss that constrains OOD features to dimensions orthogonal to the principal subspace of ID features formed by NC. Extensive experiments demonstrate state-of-the-art performance on CIFAR-10, CIFAR-100, and ImageNet benchmarks. Our work provides a foundation for further research on feature separation in OOD detection using auxiliary OOD data.

ACKNOWLEDGMENTS

The authors would like to thank the anonymous reviewers for their insightful comments.

The research leading to these results has received funding from National Key Research Development-Project (2023YFF1104202), National Natural Science Foundation of China (62376155), Shanghai Municipal Science and Technology Research Program Major Project (2021SHZDZX0102).

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

# A APPENDIX

## A.1 COMBINATION WITH OTHER OUTPUT-BASED LOSS

In our main paper, we utilize our feature separation loss based on OE method (Hendrycks et al., 2018) since it is the most classical approach. In this section, we also combine our feature separation loss with another output-based loss to demonstrate our wide availability. We adopt Energy-OE approach (Liu et al., 2020) as our basic loss, which is a commonly-used output-based loss. The mathematical formula is as follows:

$$\min_f L_{\text{CE}} + \lambda L_{\text{energy}} \tag{7}$$

$$L_{energy} = \mathbb{E}_{(x_{\text{in}},y)\sim D_{\text{in}}}(\max(0, E(x_{\text{in}}) - m_{\text{in}}))^2 \\ + \mathbb{E}_{x_{\text{out}}\sim D_{\text{out}}^{\text{aux}}}(\max(0, m_{\text{out}} - E(x_{\text{out}})))^2 \tag{8}$$

where $E(x) = -T \cdot \log \sum_i C e^{f_i(x)/T}$ is the energy score function, and $\lambda$, $m_{\text{in}}$, and $m_{\text{out}}$ are hyperparameters. We adopt the recommended setting in Energy-OE (Liu et al., 2020) to set the parameters and finetune our model. Combining our feature separation loss with the basic method, we obtain our training objective loss as follows:

$$\min_f L_{\text{CE}} + \lambda L_{\text{energy}} + \alpha L_{\text{Clu}} + \beta L_{\text{Sep}} \tag{9}$$

In our experiments, we still set $\alpha = 1.0$ and $\beta = 1.0$ for consistency with previous setting. The results are shown in Table 9, where our method significantly reduces the FPR95 by 6.35% on CIFAR100 benchmark compared to the basic Energy-OE approach, convincingly demonstrating our wide availability and effectiveness.

Table 9: Combination with Energy-OE loss on CIFAR10 and CIFAR100 benchmarks. The best result is in bold.

| Method | SVHN | | LSUN | | iSUN | | Textures | | Places365 | | Average | |
|---|---|---|---|---|---|---|---|---|---|---|---|---|
| | FPR95↓ | AUROC↑ | FPR95↓ | AUROC↑ | FPR95↓ | AUROC↑ | FPR95↓ | AUROC↑ | FPR95↓ | AUROC↑ | FPR95↓ | AUROC↑ |
| CIFAR-10 | | | | | | | | | | | | |
| Energy-OE Liu et al. (2020) | 0.75 | 99.50 | 0.90 | 98.98 | 1.50 | 99.22 | 2.75 | 98.92 | 9.05 | 97.33 | 2.99 | 98.79 |
| Ours | 1.20 | 99.33 | 0.65 | 99.14 | 1.75 | 99.33 | 2.20 | 99.08 | 7.55 | 97.88 | **2.67** | **98.95** |
| CIFAR-100 | | | | | | | | | | | | |
| Energy-OE Liu et al. (2020) | 17.75 | 96.94 | 34.00 | 94.82 | 60.75 | 87.32 | 45.70 | 90.09 | 53.50 | 89.08 | 42.34 | 91.65 |
| Ours | 11.05 | 97.65 | 21.35 | 96.40 | 52.95 | 88.63 | 42.65 | 91.14 | 51.95 | 88.81 | **35.99** | **92.53** |

## A.2 DETAILED RESULTS ON DIFFERENT ARCHITECTURES

We report the detailed results with ResNet18 and DenseNet121 architectures on CIFAR10 and CIFAR100 benchmarks in Table 10.

Table 10: Detailed Results with ResNet18 and DenseNet121 architectures. The best result is in bold.

| Model | Method | SVHN | | LSUN | | iSUN | | Textures | | Places365 | | Average | |
|---|---|---|---|---|---|---|---|---|---|---|---|---|---|
| | | FPR95↓ | AUROC↑ | FPR95↓ | AUROC↑ | FPR95↓ | AUROC↑ | FPR95↓ | AUROC↑ | FPR95↓ | AUROC↑ | FPR95↓ | AUROC↑ |
| CIFAR-10 | | | | | | | | | | | | | |
| ResNet18 | OE | 3.55 | 97.46 | 4.35 | 98.00 | 4.20 | 97.58 | 7.95 | 97.47 | 11.70 | 96.25 | 6.35 | 97.35 |
| | DAL | 0.75 | 99.38 | 1.70 | 98.94 | 2.10 | 98.05 | 4.35 | 98.20 | 9.15 | 96.45 | 3.61 | 98.20 |
| | Ours | 0.50 | 99.60 | 2.10 | 99.19 | 3.10 | 99.02 | 3.15 | 98.71 | 8.75 | 97.26 | **3.52** | **98.75** |
| DenseNet121 | OE | 4.40 | 98.51 | 4.40 | 98.68 | 22.80 | 96.19 | 5.55 | 98.52 | 16.80 | 95.78 | 10.79 | 97.54 |
| | DAL | 3.10 | 98.65 | 3.10 | 99.05 | 21.40 | 96.56 | 5.55 | 98.46 | 15.60 | 95.85 | 9.75 | 97.71 |
| | Ours | 4.45 | 98.73 | 4.45 | 98.86 | 13.90 | 97.41 | 4.90 | 98.76 | 16.80 | 95.94 | **8.90** | **97.94** |
| CIFAR-100 | | | | | | | | | | | | | |
| ResNet18 | OE | 55.75 | 92.95 | 35.45 | 93.96 | 70.10 | 87.37 | 65.00 | 88.38 | 58.50 | 88.29 | 56.96 | 90.19 |
| | DAL | 51.55 | 93.40 | 32.35 | 94.65 | 69.75 | 88.90 | 63.50 | 89.52 | 57.30 | 88.31 | 54.89 | 90.95 |
| | Ours | 36.70 | 93.29 | 34.90 | 94.12 | 66.00 | 88.68 | 57.35 | 90.01 | 56.75 | 88.39 | **50.34** | **90.90** |
| DenseNet121 | OE | 59.40 | 90.35 | 48.70 | 90.15 | 70.65 | 83.28 | 66.90 | 85.36 | 64.75 | 84.65 | 62.08 | 86.76 |
| | DAL | 47.00 | 92.81 | 60.05 | 87.88 | 55.20 | 88.65 | 65.40 | 86.90 | 78.60 | 82.07 | 61.25 | 87.66 |
| | Ours | 67.40 | 90.49 | 37.15 | 92.77 | 65.15 | 85.68 | 63.00 | 87.13 | 62.95 | 86.18 | **59.13** | **88.45** |

## A.3 HARD OOD DETECTION

In addition to testing on the regular OOD datasets, we further consider three hard OOD datasets proposed in Tack et al. (2020), which are considered more difficult to distinguish from ID samples.

Following the same setting in (Tack et al., 2020; Sun et al., 2022b; Wang et al., 2024), we evaluate our detection performance on LSUN-Fix (Yu et al., 2015), ImageNet-Resize (Deng et al., 2009b) and CIFAR100 (Krizhevsky et al., 2009a) with CIFAR10 as the ID dataset. Specific results are shown in Table 11. As we can see, our method shows comparable performance with DAL (Wang et al., 2024) over three hard OOD datasets, outperforming the baseline OE method by 2.55% on FPR95 on the ImageNet-Resize dataset.

Table 11: Hard OOD detection on CIFAR10 benchmark.

| Method | **Near-OOD Dataset** | | | | | | | |
| | LSUN-Fix | | ImageNet-Resize | | CIFAR-100 | | Tiny-ImageNet | |
| | FPR95↓ | AUROC↑ | FPR95↓ | AUROC↑ | FPR95↓ | AUROC↑ | FPR95↓ | AUROC↑ |
| | With contrastive learning | | | | | | | |
| CSI* | 39.79 | 93.63 | 37.47 | 93.93 | 45.64 | 87.64 | - | - |
| CIDER | 8.98 | 98.56 | 43.45 | 93.82 | 55.84 | 90.0 | - | - |
| KNN+* | 24.88 | 95.75 | 30.52 | 94.85 | 40.00 | 89.11 | - | - |
| | With auxiliary OOD data | | | | | | | |
| OE | 1.00 | 99.53 | 7.20 | 98.48 | 25.05 | **94.86** | 19.55 | 91.49 |
| DAL | **0.65** | **99.59** | **3.75** | **98.63** | 26.00 | 94.35 | 20.75 | 92.18 |
| Ours | 0.75 | 99.07 | 4.65 | 98.42 | **24.60** | 94.69 | **17.65** | **92.48** |

## A.4 GENERALIZATION TO CLASS-IMBALANCED DATASETS

Since our method is based on the assumption of the NC property, we recognize that the phenomenon may not always hold in real-world scenarios. Therefore, in this section, we evaluate our performance on imbalanced data distribution to explore the generalizability of our approach.

On imbalanced data distribution, the NC property declines to "Minority Collapse" Zhang et al. (2024a), where the classifiers for minority classes are squeezed into one direction. This phenomenon destroys the geometric structure of NC, but does not break the low-dimensional property of features of ID data. Following the settings in Cao et al. (2019); Zhu et al. (2022a), we create an imbalanced version of CIFAR10 dataset, denote as CIFAR10-LT. Specifically, we consider a long-tailed imbalance with ratio (denote the ratio between sample sizes of the most frequent and least frequent class). Based on above imbalanced data, we firstly pretrain a WideResNet-40-2 model on CIFAR10-LT, and then finetune the model using auxiliary OOD datasets. The training setting is the same as our previous experiments. The final detection performance of our method and OE is shown in Table 12, which validates our effectiveness under imbalanced data condition.

Table 12: Results on imbalanced CIFAR10 data. We report the FPR95/AUROC.

| Method | SVHN | | LSUN | | iSUN | | Textures | | Places365 | | Average | |
| | FPR95↓ | AUROC↑ | FPR95↓ | AUROC↑ | FPR95↓ | AUROC↑ | FPR95↓ | AUROC↑ | FPR95↓ | AUROC↑ | FPR95↓ | AUROC↑ |
| OE Hendrycks et al. (2018) | 15.15 | 96.56 | 12.00 | 97.25 | 19.70 | 96.63 | 18.05 | 96.15 | 29.75 | 93.63 | 18.93 | 96.04 |
| Ours | 14.05 | 96.38 | 12.05 | 96.93 | 13.55 | 97.38 | 24.05 | 95.63 | 24.95 | 94.07 | **17.73** | **96.08** |

## A.5 TRAINING STRATEGY CHOICE AND IMPACT

In this section, we discuss the choice of our training strategy and its impact on our performance. Our training process is consisted of two stage: Stage I-training with cross-entropy loss to ensure the occurrence of NC phenomenon; Stage II-training with all losses including cross-entropy, outlier exposure, and our proposed separation and clustering losses to enlarge the discrepancy between ID and OOD data. When a well-trained model is used as the initial parameter, Stage I is not necessary. In practice, for the sake of an unified and simple framework, we can always firstly train with CE loss, and when the accuracy remains unchanged, switch to Stage II. But in our experiment, to align with experimental settings of other methods, we fixed the training epoch at 5, which is a relatively small epoch, thus we directly use Stage II on the ImageNet benchmark.

To further investigate the impact of training strategy on our performance, we conduct experiments

on CIFAR10 benchmark (where the initial model parameter is half-trained), with different training epochs of Stage I. Specifically, denote the training epoch of Stage I as "Ep-CE", training epoch of Stage II as "Ep-All", we choose different Ep-CE ranging from [0, 10, 20, 25, 30, 40, 50] with a fixed Ep-All equalling to 25. When Ep-CE equals to zero, it corresponds to directly train the model using Stage II. Experiment results are shown in Table 13, where our method is not sensitive to the strategy choice, but CE loss fine-tuning does enhance performance for half-trained models.

Table 13: Influence of training strategy on CIFAR10 benchmark. We report the average FPR95/AUROC across five OOD datasets.

| Ep-CE + Ep-All | 0+50 | 10+25 | 20+25 | 25+25 | 30+25 | 40+25 | 50+25 |
|---|---|---|---|---|---|---|---|
| FPR95/AUROC | 2.53/98.53 | 2.56/99.06 | 2.56/99.09 | 2.49/98.93 | 2.46/99.16 | **2.01/99.11** | 2.53/98.97 |
| ID Acc | 94.43 | 95.33 | 95.52 | 95.53 | 95.56 | **95.64** | 95.35 |

### A.6 FEATURE DISTANCE ANCHORING ON CLASS MEAN

Since our training loss explicitly optimizes the cosine similarity between FC weights and features, the distance metric we used in Table 7 is align with our objective in some degree. Therefore, for a more fair comparison, we evaluate the feature distance based on the class mean vectors instead of FC weights in this section. The result is shown in Table 14. The absolute value is different from results calculated based on model weights, but in relative comparison, our method still shows better feature separation between ID and OOD data.

Table 14: Feature Distance based on class mean vectors. we report the average Euclidean distance and cosine similarity on the whole dataset.

| Method | Euclidean Distance | | | Cosine Similarity | | |
|---|---|---|---|---|---|---|
| | ID | OOD | Diff↑ | ID | OOD | Diff↑ |
| Vanilla | 2.31 | 3.08 | 0.77 | 0.94 | 0.81 | 0.13 |
| OE | 1.46 | 4.91 | 3.45 | 0.94 | 0.49 | 0.45 |
| Ours | 1.47 | 6.18 | **4.71** | 0.98 | 0.31 | **0.67** |

### A.7 FLUCTUATION IN DETECTION PERFORMANCE

We have observed considerable fluctuations in the performance of the DAL method (Wang et al., 2024) under repeated experiments with identical settings. Therefore, we evaluate the mean and variance of performance after repeating the same experiment five times. The results, shown in Table 15, indicate that our approach exhibits greater stability, particularly on the CIFAR100 benchmark. Notably, even the OE method shows the FPR95 variance of $1.137\%$ on CIFAR100, whereas our method maintains a variance of only $0.027\%$.

Table 15: Fluctuation in detection performance of different methods.

| Method | CIFAR10 | | CIFAR100 | |
|---|---|---|---|---|
| | FPR95↓ | AUROC↑ | FPR95↓ | AUROC↑ |
| OE Hendrycks et al. (2018) | $3.22_{\pm 0.0017}$ | $\mathbf{99.07}_{\pm 0.0014}$ | $36.29_{\pm 1.137}$ | $92.31_{\pm 0.013}$ |
| DAL Wang et al. (2024) | $2.87_{\pm 0.0234}$ | $98.82_{\pm 0.0027}$ | $30.44_{\pm 2.216}$ | $93.07_{\pm 0.075}$ |
| Ours | $\mathbf{2.49}_{\pm 0.0007}$ | $98.92_{\pm 0.0033}$ | $\mathbf{29.50}_{\pm 0.027}$ | $\mathbf{93.98}_{\pm 0.014}$ |

### A.8 DIFFERENT SCORE FUNCTIONS

Since we use our proposed score function in Eq 6 to detect OOD samples while other auxiliary OOD data based methods only employ MSP score (Hendrycks & Gimpel, 2016), in this part, we also evaluate our model using MSP score for a fair comparison. The results in Table 16 demonstrate

that our approach also achieves commendable performance under the MSP score, with only a slight decline compared to using the proposed score function.

Table 16: Performance of adopting different score functions in our method.

| Method | Score Function | CIFAR10 FPR95↓ | CIFAR10 AUROC↑ | CIFAR100 FPR95↓ | CIFAR100 AUROC↑ |
|---|---|---|---|---|---|
| Ours | MSP | 2.77 | 98.76 | 29.96 | 93.27 |
| | Eq. 6 | **2.49** | **98.93** | **29.47** | **94.00** |

### A.9 VISUALIZATION

In Figure 2, we visualize the random two class of samples of CIFAR10 dataset and the test unseen OOD samples of SVHN dataset. The low-dimensional visualization is calculated by linear projection into the subspace spanned by $w_1$, $w_2$ and principal components of OOD features, where $w_i$ is the corresponding model parameters of the last fully connected layer. Specifically, denote features as $z \in \mathbb{R}^{n \times d}$, the reduction matrix as $M \in \mathbb{R}^{d \times 3}$, where $n$ is the sample number and $d$ is the feature dimension, then the coordinate in 3D space is computed as: $z_{3D} = zM, z_{3D} \in \mathbb{R}^{n \times 3}$.

Using the above linear projection operation, we obtain the coordinate for both ID and OOD samples, then visualize them in 2D and 3D space. In the following, we respectively choose two class of ID samples and the test unseen OOD data to visualize their features in 2D and 3D space. In these figures, we can discover that, for all samples of the ten classes in CIFAR10, ID features rarely distribute on redundant dimensions, in contrast, OOD features almost locate on redundant dimensions while little component on model weight dimensions.

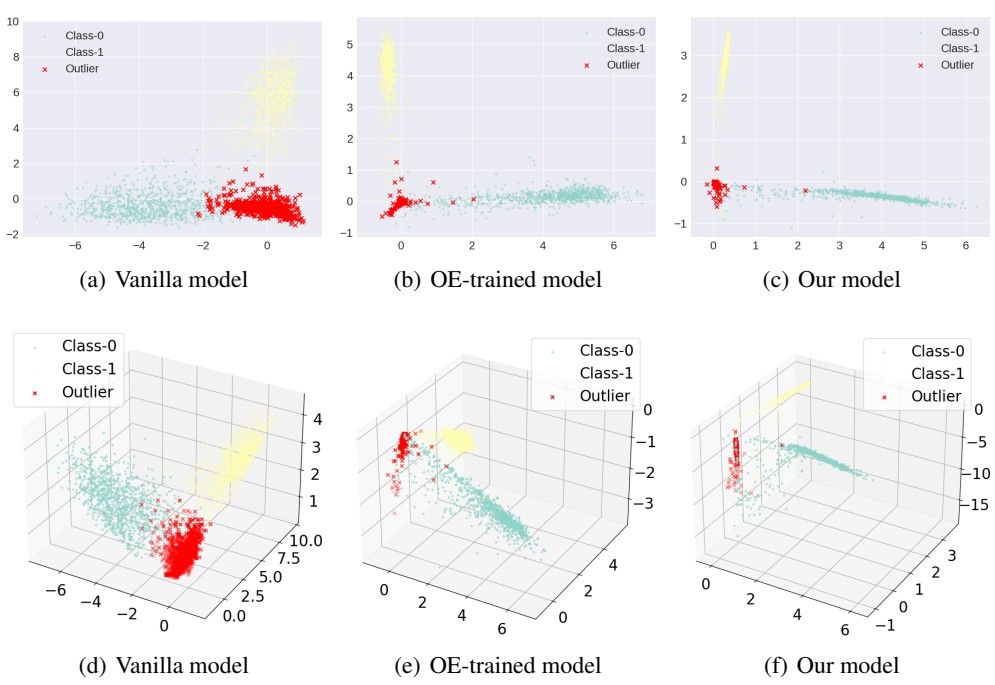

(a) Vanilla model    (b) OE-trained model    (c) Our model

(d) Vanilla model    (e) OE-trained model    (f) Our model

Figure 3: ID sample: Class-0 and Class-1, OOD sample: SVHN

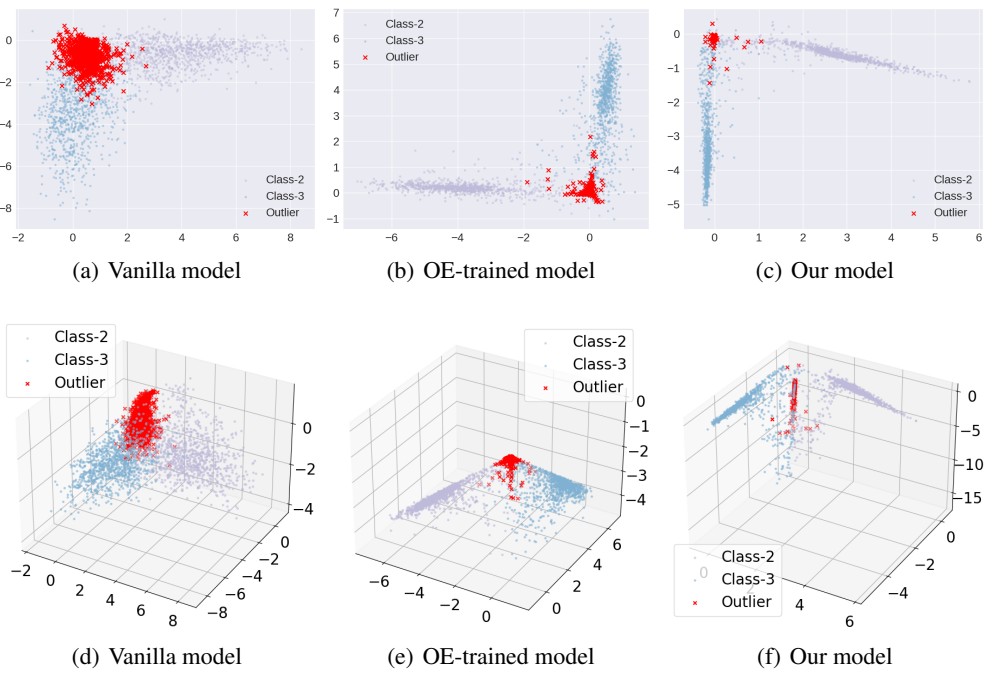

Figure 4: ID sample: Class-2 and Class-3, OOD sample: SVHN

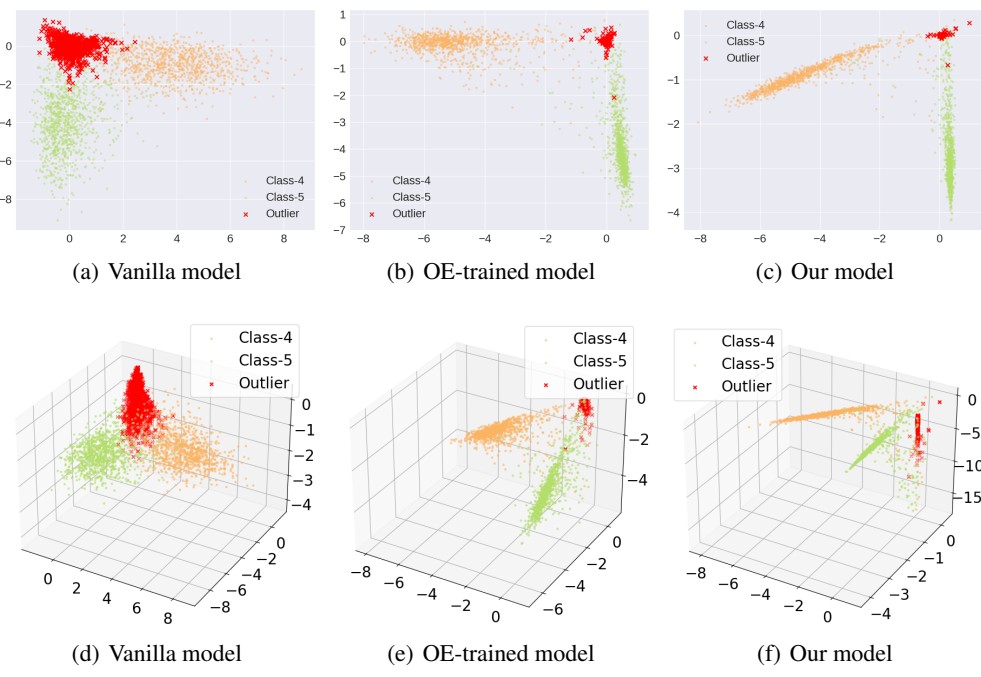

Figure 5: ID sample: Class-4 and Class-5, OOD sample: SVHN

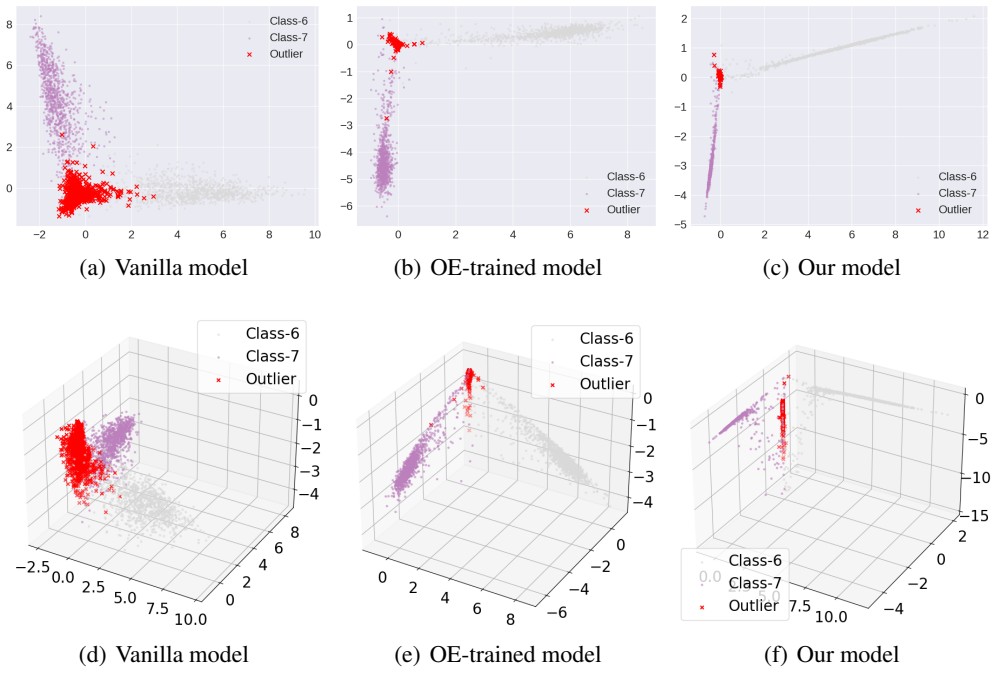

Figure 6: ID sample: Class-6 and Class-7, OOD sample: SVHN

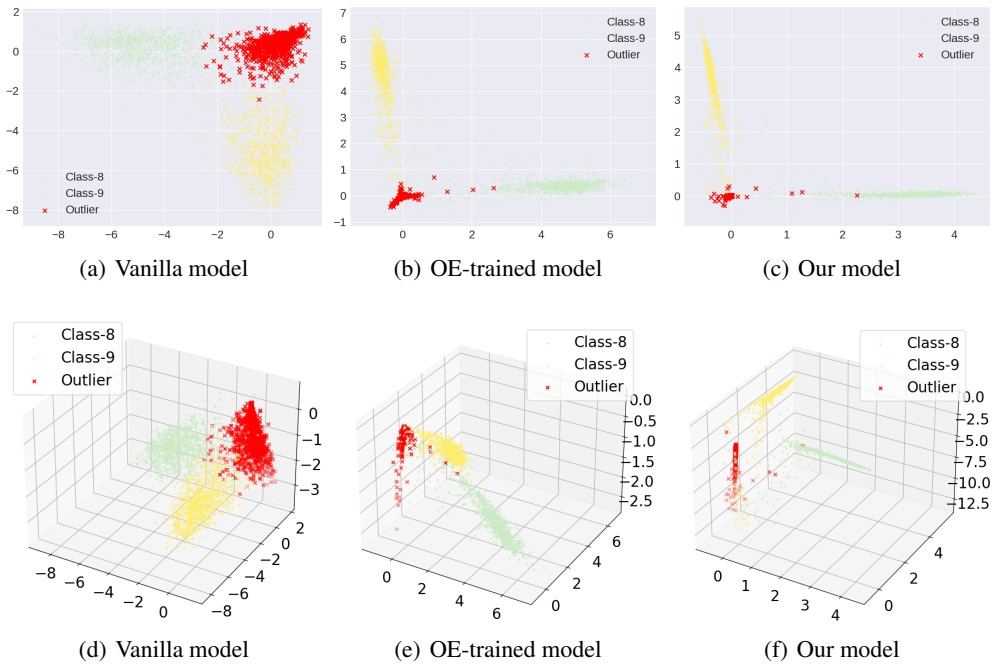

Figure 7: ID sample: Class-8 and Class-9, OOD sample: SVHN

