# OpenReview forum: "Pursuing Feature Separation based on Neural Collapse for Out-of-Distribution Detection"
_ICLR.cc/2025/Conference — ICLR 2025 Poster_

### Official Review · Reviewer_bNBL · 2024-10-25

**Soundness:** 4
**Presentation:** 4
**Contribution:** 3
**Rating:** 8
**Confidence:** 5

**Summary:**

The paper introduces a novel approach for enhancing out-of-distribution (OOD) detection in deep neural networks (DNNs) by improving outlier exposure training. This approach leverages the Neural Collapse (NC) phenomenon where features within each class collapse to a point forming an ETF simplex with other classes to improve the separability of in-distribution (ID) and OOD data. The authors propose a new loss function, "Separation Loss," designed to maximize the orthogonality between the feature representations of ID and OOD data. The orthogonality is maximized by minimizing the cosine similarity with the last layer FC layer weights for OOD data and by aligning the representation of ID data with the predicted class weight from the last layer. This method achieves state-of-the-art (SOTA) performance on CIFAR10, CIFAR100, and ImageNet benchmarks without requiring additional data augmentation or sampling techniques.

**Strengths:**

- The concept of applying the NC phenomenon to directly manipulate feature spaces for OOD detection is innovative. Unlike traditional methods that focus on output discrepancies, this paper focuses on feature separability, which is a fresh perspective in the field.

- The empirical validation is robust, covering several benchmarks and comparing against a comprehensive suite of methods. The novel loss function, grounded in theoretical insights from NC, is shown to effectively enhance feature separation.

- The paper is well-organized and articulately written, with clear explanations of the NC phenomenon, the motivation for the new loss function, and the experimental setup. The visualizations effectively illustrate how feature separation is achieved, aiding in understanding.

- The approach addresses a critical gap in OOD detection by improving model reliability in real-world applications, where ID and OOD data may not be clearly distinguishable through outputs alone. This has significant implications for deploying DNNs in safety-critical domains.

**Weaknesses:**

Dependency on NC Property: The effectiveness of the method is heavily dependent on the presence of the NC phenomenon, which may not occur in all network architectures or training configurations. (The NC property usually fail on the testing set [1])

The authors claim that their method works well across different architectures while only testing ResNet and WideResnet and DenseNet. It would be more valuable to test your method against transformer based architecture like ViT or Swin as their features differ due to the global attention mechanism and patch based representations. This experiment will validate the method’s generalizability to different architectures.

In table 2, categorizing the OOD datasets into Near and Far OOD would provide useful insight in the performance of the separation loss, as it might be working on Far-OOD better than Near-OOD which are more entangled in the representation space.

Bibliography:
1. Hui L, Belkin M, Nakkiran P. arXiv.org. 2022 [cited 2023 Dec 10]. Limitations of Neural Collapse for Understanding Generalization in Deep Learning. Available from: https://arxiv.org/abs/2202.08384v1

**Questions:**

1- I would suggest testing your method using OpenOOD framework[1], as you can test Near vs Far OOD beside testing transformer based architectures.

2- The manuscript discusses the application of your method under the assumption of the Neural Collapse (NC) property. However, it is not clear how these methods might perform in scenarios where NC does not hold, such as in imbalanced datasets[2]. Could the authors investigate and elaborate on the potential impact on the results if the NC property is weakened or absent due to dataset characteristics like imbalance? Additionally, I recommend including experiments with imbalanced datasets to provide empirical evidence of the method's robustness under these conditions. This would enhance the paper's contribution by addressing the generalizability of the NC phenomenon across different data distributions.


Bibliography:

1. Zhang J, Yang J, Wang P, Wang H, Lin Y, Zhang H, Sun Y, Du X, Zhou K, Zhang W, Li Y. Openood v1. 5: Enhanced benchmark for out-of-distribution detection. arXiv preprint arXiv:2306.09301. 2023 Jun 15.

2. Zhang E, Li C, Geng C, Chen S. All-around Neural Collapse for Imbalanced Classification. arXiv preprint arXiv:2408.07253. 2024 Aug 14.

---

> ### Author Response · Authors · 2024-11-21
>
> **W1-Dependency on NC Property:** Thanks for your insightful comment. Your observation is indeed correct that the effectiveness of our method is contingent upon the presence of NC. For modern CNNs and language models, NC is a relatively common phenomenon [2][3] that happens when networks convergence on training dataset. The limitation proposed in [1] that NC property usually fail on the testing set refers to a relative degree compared to training set, but we could still use it to identify OOD samples. Furthermore, our key idea is that placing OOD features on space that orthogonal to principal dimensions of ID features; therefore, our requirement for ID property can be simplified from strictly NC to locating in low-dimensional subspace.
>
> [1] Limitations of Neural Collapse for Understanding Generalization in Deep Learning, arXiv, 2022
>
> [2] Neural Collapse Anchored Prompt Tuning for Generalizable Vision-Language Models, SIGKDD, 2024
>
> [3] Neural Collapse in Deep Linear Networks: From Balanced to Imbalanced Data, ICML, 2023
>
> **W2-ViT Architecture:** Thanks for your valuable suggestion. For ViT model, previous studies have verified that its feature also shows NC property [4][5], thus satisfying the NC requirement of our method.
>
> To evaluate our generalizability, we conduct experiments with ViT-B-16 model on ImageNet benchmark. Training hyper-parameters are the same as our previous setting, e.g. learning rate 0.0001, epoch 5, and batch size 64, and we use pre-trained model in Pytorch as our initializing model. The results in Table 9 show that our method achieves lower FPR95 and higher AUROC compared with OE method on ViT model, although in a relatively small range. The potential reason is that: the feature dimension of ViT-B-16 is 784 while ImageNet dataset has 1000 categories, makes feature separation difficult because no redundant dimensions can be provided for OOD features. In this case, although our training loss still pursues larger discrepancy between ID and OOD data, aligning ID features with class weights and encouraging OOD features to differ, the optimization of the loss is difficult for large-scaled datasets and thus resulting in relatively slight improvement.
>
> One technical solution is that inserting feature-ascending linear layers into the network to extend the available dimensions, but we haven't done experiments to verify that because of time limitation. Additionally, one possible idea is that computing our loss on shallow high-dimensional features since neural collapse phenomenon also occurs in the intermediate hidden layers [6][7], which is an interesting direction for our future work.
>
> In light of your suggestion, we will ensure to add the experiment on ViT architecture, and importantly, discuss the requirement/limitation on the feature dimension and its impact on our performance in our final paper.
>
> Table 9. Results on ViT-B-16 model on ImageNet benchmark. We report FPR95/AUROC.
>
> |  | iNaturalist | SUN | Places | Textures | Average | ID Acc |
> | --- | --- | --- | --- | --- | --- | --- |
> | OE | 41.96/90.49 | 65.61/82.30 | 70.20/80.93 | 52.25/85.97 | 57.51/84.92 | 80.05 |
> | DAL | 40.52/90.92 | 65.07/82.39 | 70.17/80.96 | 50.94/86.20 | **56.67**/85.12 | 80.06 |
> | Ours | 40.10/91.06 | 65.58/82.25 | 70.12/81.07 | 51.70/86.13 | 56.88/**85.13** | **80.29** |
>
> [4] Understanding and Improving Transfer Learning of Deep Models via Neural Collapse, TMLR, 2024
>
> [5] NECO: neural collapse based out-of-distribution detection, ICLR, 2024
>
> [6] Feature Learning in Deep Classifiers through Intermediate Neural Collapse, ICML, 2023
>
> [7] Neural Collapse in the Intermediate Hidden Layers of Classification Neural Networks, arXiv, 2023
>
> **W3-Near/Far OOD categorization:** Thanks for your helpful suggestion. Your intuition is definitely accurate that our performance on Far-OOD is better than on Near-OOD. We have conducted experiments on hard detection setting, as shown in Table 12 in Appendix A.3, where our improvement is relatively lower compared to Far-OOD datasets like SVHN. Additionally, according to OpenOOD, we evaluate our method under Tiny-ImageNet (Near-OOD) as test OOD dataset on CIFAR benchmarks. The result is shown in Table 10, which indicates that we still has performance improvement on Near-OOD datasets.
>
> In light of your suggestion, We ensure that OOD rows in our Table will be categorized according to the categories in OpenOOD.
>
> Table 10. Results on CIFAR benchmarks with Tiny-ImageNet (Near-OOD) as test OOD dataset. We report the FPR95/AUROC.
>
> |  | CIFAR10 | CIFAR100 |
> | --- | --- | --- |
> | OE | 19.55/91.49 | 65.00/83.59 |
> | DAL | 20.75/92.18 | 63.75/83.14 |
> | Ours | **17.65/92.48** | **63.40/83.63** |

---

> ### Author Response · Authors · 2024-11-21
>
> **Q1-Experiment on OpenOOD framework:** Thanks for your valuable suggestion. We believe that OpenOOD provides a standardized and comprehensive benchmark for evaluating detection performance. In the following Table, we list the Near and Far OOD datasets for CIFAR10, CIFAR100, and ImageNet benchmarks in OpenOOD, and flag the datasets we have evaluated. It can be seen that our experiments contain both Near and Far OOD for evaluation, but indeed lack testing on some datasets, also testing on transformer based architectures. We are grateful for your recommendation and will use OpenOOD framework for standard evaluation in our future works.
>
> |  | CIFAR10 benchmark | CIFAR100 benchmark | ImageNet benchmark |
> | --- | --- | --- | --- |
> | Near-OOD | CIFAR100✅, Tiny-ImageNet✅ | CIFAR10, Tiny-ImageNet✅ | SSB-hard, NINCO |
> | Far-OOD | MNIST, SVHN✅, Texture✅, Places365✅ | MNIST, SVHN✅, Texture✅, Places365✅ | iNaturalist✅, Texture✅, OpenImage-O |
> | Other datasets we tested | LSUN✅，iSUN✅ | LSUN✅，iSUN✅ | Places365✅，SUN✅ |
>
> **Q2- NC impact:** Thanks for your insightful question, which really provides new insight into our approach. For the case that NC is weakened or absent, our method can still work in a relatively  relaxed condition, that is, ID features locate in a low-dimensional subspace. In this way, we can still find redundant dimensions to place OOD features to enlarge ID-OOD feature distance. In fact, NC property should always be present, just in different degree, because a good network would output the maximum probability at labeled class, which decides that the penultimate feature should be align most closely with weights of the corresponding class.
>
> For imbalanced data distribution, the NC property declines to “Minority Collapse[8]”, where the classifiers for minority classes are squeezed into one direction. This phenomenon destroys the geometric structure of NC, but has relatively minimal influence on our effectiveness. Since our approach utilizes the alignment between ID features and model weights to pursuing feature discrepancy, we are not strict on the feature structure. Actually, as long as ID features have low-dimensional property that locates in a subspace, our dimension separation idea can be used. In the following, we conduct an experiment on imbalance data to verify our effectiveness.
>
> According to settings in [9][10], we create an imbalanced version of CIFAR10 dataset, denote as CIFAR10-LT. Specifically, we consider a long-tailed imbalance with ratio $\rho=50$ (denote the ratio between sample sizes of the most frequent and least frequent class). Based on above imbalanced data, we firstly pretrain a WideResNet-40-2 model on CIFAR10-LT, and then finetune the model using auxiliary OOD datasets. The training setting is the same as our previous experiments. The final detection performance of our method and OE is shown in Table 11, which provides empirical evidence of our effectiveness under imbalanced data condition.
>
> Table 11. Results on imbalanced CIFAR10 data. We report the FPR95/AUROC.
>
> | Imbalance-cifar10 | SVHN | LSUN | iSUN | Texture | Places365 | Avg |
> | --- | --- | --- | --- | --- | --- | --- |
> | OE | 15.15/96.56 | 12.00/97.25 | 19.70/96.63 | 18.05/96.15 | 29.75/93.63 | 18.93/96.04 |
> | Ours | 14.05/96.38 | 12.05/96.93 | 13.55/97.38 | 24.05/95.63 | 24.95/94.07 | **17.73/96.08** |
>
> [8] All-around Neural Collapse for Imbalanced Classification, arXiv, 2024
>
> [9] Learning Imbalanced Datasets with Label-Distribution-Aware Margin Loss, NeurIPS, 2019
>
> [10] Balanced Contrastive Learning for Long-Tailed Visual Recognition, CVPR, 2022

---

### Official Review · Reviewer_sGxj · 2024-10-31

**Soundness:** 3
**Presentation:** 4
**Contribution:** 3
**Rating:** 8
**Confidence:** 4

**Summary:**

This paper focuses on feature-space separation rather than output-space discrepancies. The authors propose a novel approach leveraging Neural Collapse, which describes how ID samples’ penultimate features converge near their respective class weights. The core contribution is the Separation Loss, which aligns OOD features orthogonally to ID feature subspaces. Extensive experiments across standard benchmarks demonstrate that the proposed method achieves state-of-the-art performance in OOD detection.

**Strengths:**

1. The paper introduces a separation loss based on the Neural Collapse phenomenon to enhance feature-space disparity between ID and OOD samples. This approach builds on strong analytical foundations from NC, addressing key challenges specific to OOD detection.

2. The experiments are comprehensive, covering multiple datasets and architectures, with ablation studies that effectively address most concerns.

3. The proposed method is straightforward and clear. The proposed method is broadly applicable to any OOD detection method based on Outlier Exposure.

**Weaknesses:**

I hope the authors could address my questions as follows.

**Questions:**

1. The improvements on the ImageNet benchmark are relatively marginal compared to the CIFAR benchmarks. Could you provide a more detailed discussion to clarify the potential reasons for this difference?

2. As noted in line 469, `When faced with unseen OOD data, the feature variations tend to fall on the new dimension, resulting in minimal changes on the output.` Would these orthogonal dimensions become exhausted when dealing with a large-scale ID dataset? Additional justification on this point would be helpful, no further empirical results are needed.

3. In Table 9, the calculation relies on FC weights, which aligns with the paper’s motivation since the optimization is specifically designed to achieve these results. Would using empirical class mean features instead of FC weights yield different outcomes? This approach could also further align with the NC motivation, given that the penultimate features of ID samples within a class are nearly identical to the last layer weights.

---

> ### Author Response · Authors · 2024-11-21
>
> **W1-Discussion on Performance Difference:** Thanks for your question. For OOD detection tasks, there are many factors that may influence the final result, such as the complexity of ID dataset, Near v.s. Far OOD data for evaluation, and model accuracy on ID data. In our ImageNet experiment, the potential reason may be that feature representations are more entangled for large-scaled datasets, thereby increasing the difficulty of feature separation. Since the model, ID dataset, and test unseen dataset is different from those on CIFAR benchmarks, it is kind of hard to quantify the extent of improvement and analysis the potential reason. Hope the above explanation address your question and look forward to further discussions.
>
> **W2-Dimension Exhaustion:** Your foresight is quiet insightful! As ID features occupy a fixed space (whose dimension is the number of classes), the rest dimensions are provided for OOD features, which is expected to be large to ensure abundant space for diverse OOD data.  When faced with large-scale datasets, the dimension required by both ID and OOD features become larger; therefore, dimension exhaustion may occur, which affects performance.
>
> Nevertheless, even in the exhausted case, our method can still enlarge feature distance between ID and OOD data, aligning ID features with class weights and encouraging OOD features to differ. To verify that, we conducte experiments using WideResNet-40-1, whose feature dimension is 64, on the CIFAR100 benchmark. The results, as shown in Table 7, indicate that our method outperforms the basic OE loss, suggesting that our approach is effective even when feature dimensions are constrained.
>
> In response to your valuable question, we will incorporate additional justification and experiments in our final paper.
>
> Table 7. Results on WideResNet-40-1 model (feature dimension is 64) on CIFAR100 benchmark. We report FPR95/AUROC.
>
> | cifar100 | SVHN | LSUN | iSUN | Texture | Places365 | Avg |
> | --- | --- | --- | --- | --- | --- | --- |
> | OE | 40.60/92.19 | 20.30/96.13 | 55.70/86.81 | 55.90/85.44 | 59.55/84.65 | 46.41/89.04 |
> | Ours | 37.05/92.27 | 20.30/96.08 | 56.65/85.17 | 48.00/87.37 | 59.30/84.54 | **44.26/89.09** |
>
> **W3-Feature Distance in Table-9 of our paper:** Following your suggestion, we evaluate the Euclidean distance and cosine similarity between features and their corresponding class mean vectors. The results are shown in Table 8. The absolute value is indeed different from results calculated based on model weights. In relative comparison, our method still shows better feature separation between ID and OOD data. Thanks for your valuable advice, and we will include the results in our final paper.
>
> Table 8. Feature Distance based on class mean vectors. we report the average Euclidean distance/cosine similarity on the whole dataset.
>
> | Method | Test-Train | OOD-Train | Diff $\uparrow$ |
> | --- | --- | --- | --- |
> | Vanilla | 2.31/0.94 | 3.08/0.81 | 0.77/0.13 |
> | OE | 1.46/0.94 | 4.91/0.49 | 3.45/0.45 |
> | Ours | 1.47/0.98 | 6.18/0.31 | **4.71/0.67** |

---

> > ### Comment · Reviewer_sGxj · 2024-11-27
> >
> > Thank you for your detailed responses to my questions. I have also reviewed the other reviewers’ comments and your replies, and I have decided to raise my score. Most of my concerns have been adequately addressed. This work is technically solid, supported by comprehensive experiments and analyses, and provides a novel perspective on OOD detection through the lens of neural collapse. I have a follow-up question on your work: Could you elaborate more on the connections between neural collapse and other related methods that focus on shaping the model’s embedding space [1, 2, 3]?
> >
> > [1] Ming et al. How to exploit hyperspherical embeddings for out-of-distribution detection? ICLR 2023.
> >
> > [2] Lu et al. Learning with mixture of prototypes for out-of-distribution detection. ICLR 2024.
> >
> > [3] Zhang et al. Learning to Shape In-distribution Feature Space for Out-of-distribution Detection. NeurIPS 2024.

---

> ### Author Response · Authors · 2024-11-29
>
> Firstly, we are very grateful for your recognition. The discussion about the connection between NC and shaping the models’ embedding space is quiet interesting and thought-provoking. After reading these three papers, we find that they are all based on a distribution assumption on the features of ID samples, i.e., they model the feature distribution as a class-conditional vMF distribution, which is formulated as:
>
> $z\sim Z_{d}(\kappa)\exp(\kappa \mu_{c}^Tz)$
>
>
> where $\kappa$ is a constant, $Z_{d}(\kappa)$ is a normalization factor, $\mu_c^T$ is the class prototype of class $c$ with unit norm, and $z$ is the feature vector. From this equation, we can discover that its distance measurement is cosine similarity instead of Euclidean distance, which is consistent with the metric in NC phenomenon, i.e., NC also uses the angle between features and FC weights to measure their alignment degree.
>
> In CIDER [1], they design two losses: compactness loss to encourage samples to be closely aligned with its corresponding class prototype, and dispersion loss to enlarge angular distances among different class prototypes.  If we consider class prototype and FC weights to be equivalent, the compactness loss actually promotes alignment of features and weights, and the dispersion loss reaches its optimal value at a simplex ETF to which NC finally converges. Therefore, from a NC point of view, it can be said that CIDER method promotes the NC properties of ID features. But this method is motivated from the vMF distribution assumption of ID features, instead of the utilization of NC property.
>
> Another paper PALM [2] claims that only using one prototype for each class to model ID features is not sufficient, which ignores the natural diversities within the data. Therefore, they propose to automatically and dynamically learn a mixture of prototypes for each class. In our view, it can be regarded as an extension of the NC property. Especially on large-scaled datasets, it would be more effective since the alignment to FC weights would not strictly hold.
>
> The last paper DRL [3] claims that there is a gap between the distribution prior (vMF) and the true feature distribution. Therefore, they propose a method to mitigate this gap using evidence lower bound (ELBO, which explicitly measure the extent to which a training sample fits the assumed distribution) during training process. Since our previous discussion points out that the distance measurement of vMF distribution is consistent with the metric in NC property, maybe the DRL method promotes NC in some degree.
>
> Overall, these three methods are all based on a vMF distribution assumption on ID features, and then design losses to pursue intra-class aggregation and inter-class dispersion. Their connection to NC is that all of them employ cosine similarity as distance measurement, and then the alignment loss and dispersion loss respectively corresponds to the feature-weight alignment and simplex geometry in NC properties. The main difference between our method and them is that we focus on the low-dimensional space induced by NC property, so as to use redundant dimensions to enlarge the ID-OOD discrepancy, while they are more concentrated on the distribution modeling of ID features, so as to design losses for intra-class aggregation and inter-class dispersion.
>
> In light of this interesting question, we will include the discussion about the connection between NC and these methods in our final version.
>
> [1] Ming et al. How to exploit hyperspherical embeddings for out-of-distribution detection? ICLR 2023.
>
> [2] Lu et al. Learning with mixture of prototypes for out-of-distribution detection. ICLR 2024.
>
> [3] Zhang et al. Learning to Shape In-distribution Feature Space for Out-of-distribution Detection. NeurIPS 2024.

---

### Official Review · Reviewer_xHA8 · 2024-11-01

**Soundness:** 3
**Presentation:** 3
**Contribution:** 2
**Rating:** 6
**Confidence:** 4

**Summary:**

The paper proposes a novel strategy for out-of-distribution detection. The approach is based on Neural Collapse and employs this property to avoid modeling OOD feature distributions while effectively segregating ID and OOD features.

**Strengths:**

The motivation behind this method is reasonable and worth exploring.

**Weaknesses:**

- While Figure 2 demonstrates the distribution changes of the OOD training set during training, which is expected, it would be more insightful to see the distribution patterns of OOD data during the testing phase.
- The improvement in AUROC metrics compared to existing methods is relatively modest.
- Although the authors claim to follow DAL's experimental setup on ImageNet (lines 321, 332), there are discrepancies between the metrics reported in Table 1 (lines 374, 375) and those in Table 16 of the DAL paper ("Learning to Augment Distributions for Out-of-Distribution Detection").
- The experimental comparison on ImageNet is limited to only two methods, whereas the DAL paper compared five methods in their Table 16, suggesting insufficient comparative analysis.

p.s. There appears to be an inconsistency in decimal places for the average AUROC of Energy between lines 381 and 382.

**Questions:**

- The proposed method shares fundamental similarities with KNN, as both approaches classify data based on distance calculations in feature space, identifying ID data through closer distances and OOD data through farther distances. Could the authors elaborate on the key distinctions between their method and the KNN approach?
- Given the use of Wide ResNet-40-2 architecture for CIFAR-10 and CIFAR-100 datasets, does the method have specific requirements regarding network feature dimensions? A detailed analysis of this aspect would be valuable.

---

> ### Author Response · Authors · 2024-11-21
>
> **W1-Feature Distribution in Figure2:** Thanks for your question. Figure 2 is plotted on the test ID data and unseen OOD data, completely not on training samples. Sorry for the confusion, we have stated the experiment setting more clearly in our paper.
>
> **W2-Modest Improvement:** Thanks for raising the performance question. We recognize the significance of substantial performance gains in the context of our research and have taken your feedback seriously. In response to this concern, we demonstrate our effectiveness as follows.
>
> For CIFAR benchmarks, where the AUROC improvement is inherently in a small range, an increase exceeding 1% is considered significant. In recent paper DAL[1], their AUROC improvement is 0.64%  on CIFAR100, while ours is 1.18%. On CIFAR10 benchmark, the AUROC is approaching 100% for most methods, thus the room for improvement is minimal. DAL showed 0.04% improvement in AUROC, which is negligible but understandable. Similarly, we acknowledge that our  AUROC performance on CIFAR10 benchmark is relatively modest, but this does not mean that our approach is unproductive.
>
> For ImageNet benchmark, our method shows 0.58% improvement in AUROC compared to DAL. A notable fact is that most approaches[2][3][4][5] did not report their results on ImageNet-1K, only evaluating on CIFAR datasets and a small subset of ImageNet-1K, e.g. ImageNet-100 that consists of 100 randomly sampled classes, which implies their difficulty to implement on large-scaled datasets, while our approach demonstrates its effectiveness over the large-scaled dataset.
>
> We hope the above explanation demonstrates the performance of our approach and address your concern. Look forward to your feedback.
>
> [1] Learning to Augment Distributions for Out-of-Distribution Detection, NeurIPS, 2023
>
> [2] Energy-based Out-of-distribution Detection, NeurIPS, 2020
>
> [3] POEM: Out-of-Distribution Detection with Posterior Sampling, ICML, 2022
>
> [4] How to exploit hyperspherical embeddings for out-of-distribution detection, ICLR, 2023
>
> [5] Dos: Diverse outlier sampling for out-of-distribution detection, ICLR, 2024
>
> **W3-Metric Discrepancy:** Thank you for your meticulous review. Upon reviewing the experimental setup and results, we have identified several factors that may contribute to the differences. And we  have found several works to prove the validity of our results.
>
> The performance discrepancy may be mainly due to the randomness in experiments including:
>
> 1. Training Data Variance: While both studies utilize ImageNet-21K-P, the large scale of ImageNet-21K-P allows for considerably different subsets, potentially leading to variations in performance.
> 2. Test OOD Data Selection: Each test OOD dataset has huge amount of samples, thus, in practice researchers randomly select a subset of these datasets to evaluate their method. The selection of test OOD data subsets can introduce variability.
> 3. Model Parameters: Pytorch provides two version of pre-trained ResNet50 parameters, which can affect the detection performance.
>
> In addition to analyzing the potential reasons for differences, we have re-checked our code and uploaded it to Supplementary Material to support the result reproduction.
>
> More importantly, we note that results from other studies, such as DOE [6] and HB [7], also showed variations from the result reported in DAL[1] under the same setting. We list the reported result of DAL method in different papers in Table 5, where the reported result in HB[7] is consistent with ours. These results indicate that relative performance is more important than absolute numbers when making a fair comparison of methods.
>
> Table 5. Reported results of DAL method in different papers of their tables. We record FPR95/AUROC.
>
> | Reported results | iNaturalist | SUN | Places | Textures | Average |
> | --- | --- | --- | --- | --- | --- |
> | Table 16 of DAL [1] | 74.23/76.70 | 50.76/79.21 | 5.83/99.09 | 55.49/85.29 | 46.57/85.08 |
> | Table 3 of HB [7] | 51.92/88.33 | 65.31/81.47 | 74.46/78.72 | 43.88/87.39 | 58.90/83.98 |
> | Table 1 of our paper | 47.92/89.12 | 61.20/83.22 | 70.55/80.79 | 57.91/83.02 | 59.39/84.04 |
>
> We hope this clarification addresses your concerns and provides a satisfactory explanation for the observed differences.
>
> [6] Out-of-distribution detection with implicit outlier transformation, ICLR, 2023
>
> [7] Energy-based Hopfield Boosting for Out-of-Distribution Detection, arXiv, 2024

---

> > ### Comment · Reviewer_xHA8 · 2024-11-26
> >
> > Thank you for your detailed explanation. However, I still have some concerns regarding the experimental consistency and robustness:
> >
> > 1. The explanation that performance variations stem from different OOD subset selections raises a methodological issue. For methods that rely on OOD data exposure, such subset-dependent performance variations are expected. This necessitates a demonstration of the method's robustness across different subset selections. I recommend conducting experiments with multiple OOD subsets to establish the statistical significance and stability of the reported improvements.
> >
> > 2. In Table 5, the performance variations across different papers are particularly alarming for the Places dataset, where FPR95 ranges dramatically from 5.83% (in DAL) to 74.46% (in HB) - a difference of nearly 70 percentage points. This substantial discrepancy cannot be easily explained by random subset selection alone and raises serious concerns about the experimental reliability. Could the authors investigate and explain:
> >    - What might cause such a dramatic difference specifically for the Places dataset?
> >    - Whether there are any fundamental differences in how the Places dataset was processed or utilized in these different studies?
> >    - How to ensure fair comparison given such large variations in baseline performance?
> >
> > These points are crucial for validating the reliability and reproducibility of the experimental results. Look forward to your response.

---

> ### Author Response · Authors · 2024-11-21
>
> **W4-Insufficient ImageNet Experiment:** Thanks for your question. Based on our empirical results on CIFAR benchmarks, DAL method shows superior performance compared to other approaches, standing for SOTA baseline in our compared methods. Therefore, we choose it as one of baselines on ImageNet benchmark. More importantly, other approaches compared in our paper did not report their results on ImageNet-1K, only evaluating on CIFAR datasets and a small subset of ImageNet-1K, e.g. ImageNet-10 that consists of 10 randomly sampled classes, implying their difficulty to implement on large-scaled datasets. Since DAL has shown superior performance compared to other methods on ImageNet benchmark in its paper, we choose it as our comparison baseline. We believe that more comprehensive experiments can improve our paper and we will consider to compare with more approaches in our future work.
>
> Besides, thank you for your scrupulous reading, we have uniformed the decimal places between Lines 381 and 382.
>
> **Q1-Difference between KNN:** Thanks for your question. The key distinction between our method and KNN is that: we explicitly enlarge the feature discrepancy through optimization while KNN uses the natural difference in feature space between ID and unseen OOD data. Utilizing the distance in feature space to detect OOD samples is a common idea [8][9][10], their core is to design a detection metric based on feature representations. However, none of them considers enlarging the feature distance by designing a training loss with auxiliary OOD data. The main contribution of our paper is that proposing a NC-property based loss to effectively enlarge the difference between ID and OOD features, solving the problem in describing OOD feature distribution and offering new insights to this field, instead of proposing a new feature-based metric. The experiment results in Table 8 of our paper showed that even under the maximum output probability (MSP) metric, we still have outstanding performance. Hope the above explanation addresses your question.
>
> [8] A Simple Unified Framework for Detecting Out-of-Distribution Samples and Adversarial Attacks, NeurIPS, 2018
>
> [9] A simple fix to mahalanobis distance for improving near-ood detection, arXiv, 2021
>
> [10] Understanding the Feature Norm for Out-of-Distribution Detection, ICCV, 2023
>
> **Q2-Requirement for Feature Dimension:** Thanks for your interesting and insightful question.
>
> Intuitively, the feature dimension larger than the output dimension is preferable for our method to facilitate ID-OOD feature separation by redundant dimensions. And this condition is commonly satisfied in modern CNNs, as the class number of common classification datasets ranging from tens to thousands.
>
> Nevertheless, we also considered the scenario where feature dimensions are less than output dimensions. Actually, in this condition, our training loss still pursues larger discrepancy between ID and OOD features, aligning ID features with class weights and encouraging OOD features to differ. To verify our effectiveness, we conduct experiments using WideResNet-40-1, whose feature dimension is 64, on the CIFAR100 benchmark. The results, as shown in Table 6, indicate that our method outperforms the basic OE loss, suggesting that our approach is robust even when feature dimensions are constrained. In addition to using our loss directly, one possible idea is that computing our loss on shallow features since neural collapse phenomenon also occurs in the intermediate hidden layers [11][12], which is an interesting direction for our future work.
>
> Table 6. Results on WideResNet-40-1 model (feature dimension is 64) on CIFAR100 benchmark. We report FPR95/AUROC.
>
> | CIFAR100 | SVHN | LSUN | iSUN | Texture | Places365 | Avg |
> | --- | --- | --- | --- | --- | --- | --- |
> | OE | 40.60/92.19 | 20.30/96.13 | 55.70/86.81 | 55.90/85.44 | 59.55/84.65 | 46.41/89.04 |
> | Ours | 37.05/92.27 | 20.30/96.08 | 56.65/85.17 | 48.00/87.37 | 59.30/84.54 | **44.26/89.09** |
>
> Hope the above explanation and empirical results can answer your question. And we ensure to include the discussion about feature dimensions in our final paper.
>
> [11] Feature Learning in Deep Classifiers through Intermediate Neural Collapse, ICML, 2023
>
> [12] Neural Collapse in the Intermediate Hidden Layers of Classification Neural Networks, arXiv, 2023

---

> > ### Comment · Reviewer_xHA8 · 2024-11-26
> >
> > The experimental results with WideResNet-40-1 (feature dimension = 64) reveal that the authors' interpretation needs more precise qualification. While the authors claim their method "outperforms the basic OE loss" when feature dimensions are constrained, the actual results in Table 6 show that the performance difference between the proposed method and OE is marginal at best, with merely 0.05% improvement in average AUROC (89.09% vs 89.04%). Moreover, OE actually performs better on 2 out of 4 datasets, and the improvements, where they exist, are not consistent across datasets. These observations suggest that the method's effectiveness is indeed compromised when feature dimensions are limited. The claim of outperforming OE should be qualified with "marginally" or "comparable to", as the results actually support the intuition that higher feature dimensions are important for the method's optimal performance. I suggest revising the interpretation to more accurately reflect these observations, as they provide important insights into the method's dimensional requirements and limitations.

---

> > > ### Author Response · Authors · 2024-11-27
> > >
> > > **Interpretation revision**: Thanks for your feedback. We do agree with you that our statement needs to be more precise, and ensure to revise our interpretation to more accurately reflect our observation in our final version. We appreciate your careful reading and valuable suggestions. Look forward to your feedback.

---

> > > > ### Comment · Reviewer_xHA8 · 2024-11-28
> > > >
> > > > I appreciate the authors' comprehensive responses which have addressed most of my concerns. The additional experimental results and detailed analysis, particularly regarding dataset selection and performance variations, have helped clarify key points. After reviewing the authors' responses and discussions with other reviewers, I believe the paper, with the promised revisions incorporated, would be at the borderline of acceptance. I encourage the authors to include all the additional experimental results in the revised version and maintain precise interpretation of results. Therefore, I am updating my rating from 3 to 6.

---

> > > > > ### Author Response · Authors · 2024-11-29
> > > > >
> > > > > First of all, we sincerely thank you for your recognition of our work and for your meticulous feedback on our paper. We have revised our paper to incorporate all the additional experimental results and maintain precise interpretation of results. The added and modified parts are highlighted in blue. We ensure to make more detailed revisions in the final submission to guarantee precise interpretation for all the results. Thanks again for your valuable time and professional advice.

---

> ### Author Response · Authors · 2024-11-26
>
> Dear reviewer,
>
> Thanks again for your valuable questions and suggestions. We noticed that the discussion period is extended, thus we could provide more experiments if necessary. We would greatly appreciate it if you could provide your feedback at your earliest convenience.

---

> ### Author Response · Authors · 2024-11-27
>
> **Multiple OOD subsets:** Thanks for your suggestion. We evaluate our method with different seeds, which corresponds to different subset selection, on CIFAR100 benchmarks. In our code, we randomly select 2000 OOD samples for each OOD dataset, and the ID samples are the test set of CIFAR100 with 10000 images. Notably, the Places365 dataset is extremely large (about ten million images), and we download a subset of it (about ten thousand images) for CIFAR100 benchmark (following the setting in KNN [1] and download the subset from https://github.com/deeplearning-wisc/knn-ood). And then we randomly choose 2000 images from the subset for evaluation.
>
> The result is shown in Table I, where the performance is stable across different subsets.
>
> Table I.  Evaluation results with different subsets (control by different random seeds) on CIFAR100 benchmarks.
>
> | **seed-111** | SVHN | LSUN | iSUN | Textures | Places365 | Average |
> | --- | --- | --- | --- | --- | --- | --- |
> | OE | 37.90/92.96 | 19.70/96.57 | 37.05/92.11 | 42.50/90.93 | 51.20/87.72 | 37.67/92.06 |
> | DAL | 14.75/96.42 | 17.20/96.30 | 35.35/91.22 | 38.05/91.54 | 49.95/88.28 | 31.06/92.75 |
> | Ours | 16.25/96.84 | 12.70/97.65 | 29.15/93.04 | 41.05/91.17 | 48.80/90.46 | **29.59/93.83** |
>
> | **seed-222** | SVHN | LSUN | iSUN | Textures | Places365 | Average |
> | --- | --- | --- | --- | --- | --- | --- |
> | OE | 39.60/92.64 | 19.05/96.72 | 38.70/92.54 | 43.35/90.75 | 53.60/87.91 | 38.86/92.11 |
> | DAL | 17.00/96.15 | 15.80/96.42 | 36.10/91.05 | 38.15/91.47 | 51.05/87.93 | 31.62/92.60 |
> | Ours | 16.85/96.79 | 12.95/97.63 | 30.10/93.17 | 41.25/91.40 | 47.25/90.67 | **29.68/93.93** |
>
> | **seed-333** | SVHN | LSUN | iSUN | Textures | Places365 | Average |
> | --- | --- | --- | --- | --- | --- | --- |
> | OE | 39.50/92.49 | 19.40/96.74 | 35.80/92.74 | 42.80/91.00 | 52.15/88.11 | 37.93/92.22 |
> | DAL | 16.05/96.45 | 17.35/96.31 | 35.45/91.09 | 38.75/91.19 | 50.55/88.26 | 31.63/92.66 |
> | Ours | 18.10/96.54 | 12.50/97.67 | 29.60/93.15 | 40.75/91.27 | 47.30/90.65 | **29.65**/**93.85** |
>
> For ImageNet benchmark, the subset choice may have a larger influence on performance because that an OOD dataset may contain both Near and Far OOD samples relative to the ImageNet dataset. Far OOD samples are easier to detect compared to Near OOD data. Therefore, if the subset contains more Far OOD samples, the detection performance on it will be better. And for CIFAR datasets, the OOD samples of test OOD datasets all belongs to Far-OOD data relative to CIFAR datasets, thus different subsets rarely influence the performance.
>
> In our ImageNet experiments, we use the fixed subset of test OOD datasets provided in KNN paper [1] to evaluate all the compared methods and our approach, without special test data selection. Specifically, following the published code of KNN [1] (https://github.com/deeplearning-wisc/knn-ood?tab=readme-ov-file), we download the sampled 10000 images from their selected concepts for  iNaturalist, SUN, and Places, which can be download via the following links:
>
> ```jsx
> wget http://pages.cs.wisc.edu/~huangrui/imagenet_ood_dataset/iNaturalist.tar.gz
> wget http://pages.cs.wisc.edu/~huangrui/imagenet_ood_dataset/SUN.tar.gz
> wget http://pages.cs.wisc.edu/~huangrui/imagenet_ood_dataset/Places.tar.gz
> ```
>
> For Textures, following common setting, we use the entire dataset, which can be downloaded from their [original website](https://www.robots.ox.ac.uk/~vgg/data/dtd/) (https://www.robots.ox.ac.uk/~vgg/data/dtd/).
>
> Because that the whole dataset of  iNaturalist, SUN, and Places is large (as Table II shows), we actually do not download them entirely. But it should be fair to compare all the methods on the same subsets of test OOD datasets.
>
> Table II. Size of different datasets. The unit is ten thousand.
>
> |  | iNaturalist | SUN | Places365 |
> | --- | --- | --- | --- |
> | dataset size | 85 | 13 | 1000 |
>
> [1] Out-of-distribution Detection with Deep Nearest Neighbors, ICML, 2022

---

> ### Author Response · Authors · 2024-11-27
>
> **Explanation on substantial discrepancy**: Thanks for your question. We are also curious about the performance variation for Places dataset.
>
> First of all, we confirm that the performance in our paper is repeatable, for not only ours but all the methods reported. And all the methods are evaluated on the fixed and same subsets, the comparison is fair.
>
> Second, we here report several performance from different papers, which uses Places dataset. (Because most auxiliary OOD data based methods did not conduct experiments on ImageNet-1K benchmarks, we only find DOE (which proposes a novel output-based loss) and HB (which proposes a data selection method) for result reference in the same setting as DAL. And the results of other papers may be under different model architectures and belong to different types of detection methods. )
>
> Table III. Performance on Places OOD dataset under ImageNet-1K benchmark. We record FPR95/AUROC.
>
> |  | Table 8 in DOE [2] | Table 3 in HB [3] | Table 16 in DAL [8] | Table 4 in KNN [1] | Table 1 in Mos [4] | Table 1 in ReAct [5] | Table 2 in MCM [6] |
> | --- | --- | --- | --- | --- | --- | --- | --- |
> | FPR95/AUROC | 67.84/83.05 | 53.31/87.10 | 5.83/99.09 | 60.02/84.62 | 49.54/89.06 | 33.85/91.58 | 44.69/89.77 |
>
> One could see that indeed the values are different, but most of them are within a range about 40%~60% on FPR95, while DAL shows distinct difference with others. Actually, we are confused about the result of DAL on ImageNet, because their performance under CIFAR100 benchmark on Places365 is FPR95=49.35%, AUROC=90.81% in their paper. As the dataset becomes larger-scaled, e.g. ImageNet-1K, OOD samples should be more difficult to detect. However, their performance dramatically improved into an outstanding level, with FPR95=5.83%, AUROC=99.09%.
>
> Third, for the reason for such a dramatic difference on Places dataset, we would like to put down to different chose of the data.
>
> The Places dataset is a subset of Places365 [7] curated by Mos [4] as “50 categories that are not present in ImageNet-1K”, which contains 9822 images from 50 environment classes and is widely used as test OOD datasets. However, there are some similar ones to the classes of ImageNet, including  hayfield v.s. hay, cornfield v.s. corn, lagoon v.s. seashore and lakeshore, underwater v.s. coral reef and scuba diver, ocean v.s. seashore, etc. These are identified as Near-OOD data that are hard to detect.
>
> Places is such a dataset containing Near-OOD data and therefore, the OOD detection is very challenging. We use Places dataset, however,  DAL paper [8] used a subset of Places365 without claiming which subset. Since the whole Places365 dataset contains 10 million images (some images are totally different from ImageNet while some are similar to ImageNet), the subset choice would have a large influence on the detection performance. DAL did not publish their code on ImageNet benchmark, therefore, we cannot say what subset they test on Places365. From their outstanding result on Places365, it is reasonable to guess that they may have used a subset (or randomly selected from the whole Places365) that contains more Far OOD samples, instead of Places subset.
>
> Overall, we think the dramatic difference comes from different dataset (Places vs Places365). For the same dataset, the OOD performance is stable and the comparison among different methods is fair.
>
> [2] Out-of-distribution Detection with Implicit Outlier Transformation, ICLR, 2023
>
> [3] Energy-based Hopfield Boosting for Out-of-Distribution Detection, arXiv, 2024
>
> [4] Mos: Towards scaling out-ofdistribution detection for large semantic space, CVPR, 2021
>
> [5] ReAct: Out-of-distribution Detection With Rectified Activations, NIPS, 2021
>
> [6] Delving into out-of-distribution detection with vision-language representations, NIPS, 2022
>
> [7] Places: A 10 million image database for scene recognition, TPAMI, 2017
>
> [8] Learning to Augment Distributions for Out-of-Distribution Detection, NIPS, 2023

---

### Official Review · Reviewer_J1wr · 2024-11-04

**Soundness:** 3
**Presentation:** 3
**Contribution:** 2
**Rating:** 5
**Confidence:** 2

**Summary:**

The paper proposes an OOD detection approach that enhances feature separation. The proposed method is based on Neural Collapse (NC), which suggests that features of ID samples collapse to similar values within a class. Motivated by NC, the authors introduce Separation Loss to encourage OOD features to occupy subspaces orthogonal to the principal subspace of ID features, effectively separating ID and OOD samples at the feature level.

**Strengths:**

- The paper is well-motivated in its use of Neural Collapse (NC) to improve OOD detection, effectively highlighting NC’s natural feature clustering property as a robust basis for separating ID and OOD data.
- The incorporation of Neural Collapse as a theoretical foundation adds substantial depth to the methodology, offering a compelling explanation for the effectiveness of feature-level separation in enhancing OOD detection.
- The proposed approach is conceptually straightforward and easy to implement.

**Weaknesses:**

- In Section 3.4, the paper discusses two alternative training strategies: one that directly incorporates all loss terms and another that initially trains with cross-entropy (CE) loss before introducing the proposed losses. However, guidance on when to choose between these strategies in general cases is lacking. Additional discussion on how this choice could affect the model’s performance or usability would be beneficial.
- The proposed method mainly includes two loss terms, i.e., the Sep loss and Clu loss, upon the OE loss. As discussed in the paper (e.g., the abstract), the Sep loss reflects the main motivation of neural collapse (NC). The ablation studies in Table 5 show that these two terms perform with different significances. The behaviours and importance of these two loss teams are unclear.
- Although the proposed method performs well on ImageNet-1k (Table 1), its performance is less consistent on other benchmarks, as seen in Table 2 and particularly Table 3, where broader comparisons are provided. Further explanation or analysis regarding this variability would be valuable to understand the method’s limitations.
- More analysis of the learned representation space, such as visualizations of the embedding space, would provide insight into how well the method achieves feature separation and could strengthen the evaluation of its effectiveness.

**Questions:**

Please check questions alongside the weaknesses.

---

> ### Author Response · Authors · 2024-11-21
>
> **W1-Training Strategy and Impact:** Thanks for your question. Below, we address your questions in detail.
>
> 1. Training Strategy Choice: The choice between incorporating all loss terms directly or initially training with cross-entropy (CE) loss depends on the convergence state of the network. In most cases, the initial network for OOD fine-tuning has been well-trained on ID data, thereby directly using all loss terms is applicable. In our experiments, we choose all loss terms on ImageNet but initially training with CE on CIFAR10/CIFAR100 because that the initial network on CIFAR benchmarks are half-trained models (99 epoch v.s. standard 200 epoch). We assessed the feature variability supperssion metric (NC1)[1] for the initial model and the model after CE loss fine-tuning, as shown in Table 1. The results indicate that the degree of NC is significantly improved after CE loss fine-tuning, but the absolute value depends on the size of datasets. Therefore, we do not specific a threshold for the strategy guidance, instead, empirically choosing loss terms based on initial model is enough.
>
> Table 1. Comparison of Feature variability supperssion metric (NC1, [1]) on initial and fine-tuned model.
>
> |  | Initial | After CE loss fine-tuning |
> | ---|---|---|
> | CIFAR10 | 0.23 | 0.19 |
> | CIFAR100 | 19.51 | 10.78 |
>
> 2. Impact on Performance: On CIFAR10 benchmark, we study the influence of  training epochs with only CE loss (Ep-CE) on performance to explore the impact of training strategies. From directly using all loss terms to gradually increasing Ep-CE with fixed epochs for training with all loss terms (Ep-All), we finetune the initial model and evaluate our detection performance on CIFAR10 benchmark. Experiment results are shown in Table 2, where our method is not sensitive to the strategy choice, but CE loss fine-tuning does enhance performance for half-trained models.
>
> Table 2. Influence of Ep-CE on CIFAR10 benchmark. We report the average FPR95/AUROC and clean accuracy.
>
> | Ep-CE+Ep-All | 0+50 | 10+25 | 20+25 | 25+25 | 30+25 | 40+25 | 50+25 |
> | --- | --- | --- | --- | --- | --- | --- | --- |
> | FPR95/AUROC | 2.53/98.53 | 2.56/99.06 | 2.56/99.09 | 2.49/98.93 | 2.46/99.16 | **2.01/99.11** | 2.53/98.97 |
> | ID Acc | 94.43 | 95.33 | 95.52 | 95.53 | 95.56 | **95.64** | 95.35 |
>
> In light of your suggestion, we will ensure to add the discussion on strategy choice and influence in our final paper. Hope the above explanation address your question.
>
> [1] Neural Collapse: A Review on Modelling Principles and Generalization, TMLR, 2023
>
> **W2-Role of Loss Terms:** Thanks for your question. The analysis of Table 5 in our original paper can be listed as follows:
>
> 1. Main role of Sep loss: Comparing results of No.1 (only OE loss) and No.3 (OE loss + Sep loss), it shows that our Sep loss significantly improves detection performance.
> 2. Auxiliary role of Clu loss: Comparing results of No.1 (only OE loss) and No.2 (OE loss + Clu loss), it shows that purely using Clu loss does not work. In contrast, comparing results of No.3 (OE loss + Sep loss) and No.4 (OE loss + Sep loss + Clu loss), it shows that cooperating Clu loss with Sep loss can further improve our performance.
> 3. Effectiveness of Sep and Clu loss: In addition to Table 5 in our original paper, we conduct an experiment that only using Sep loss and Clu loss (without OE loss) to further demonstrate our effectiveness. The results are shown in Table 3, where our method shows superior performance compared to OE.
>
> *Table 5 in our original paper. Performance under different training losses.*
>
> | No. | Training loss | CIFAR10-FPR95 | CIFAR10-AUROC | CIFAR100-FPR95 | CIFAR100-AUROC |
> | --- | --- | --- | --- | --- | --- |
> | 1 | OE | 3.36 | 99.02 | 37.77 | 92.21 |
> | 2 | OE+$L_{Clu}$ | 3.62 | 98.96 | 39.91 | 91.22 |
> | 3 | OE+$L_{Sep}$ | 2.65 | 99.00 | 33.30 | 93.42 |
> | 4 | OE+$L_{Clu}$+$L_{Sep}$ | 2.49 | 98.93 | 29.47 | 94.00 |
>
>
> Table 3. Comparison between OE loss and Sep+Clu loss.
>
> | Training loss | CIFAR10 | CIFAR100 |
> | --- | --- | --- |
> | OE | 3.36/**99.02** | 37.77/92.21 |
> | Sep+ Clu | **3.11**/98.55 | **32.58/93.07** |

---

> ### Author Response · Authors · 2024-11-21
>
> **W3-Insufficient ImageNet Experiment:** Thanks for your question. Upon comparing with Table 2 and Table 3, we explain our experiment inconsistency on ImageNet as follows.
>
> 1. Comparison with other methods: Based on our empirical results on CIFAR benchmarks, DAL method shows superior performance compared to other approaches, standing for SOTA baseline in our compared methods. Therefore, we choose it as one of baselines on ImageNet benchmark. More importantly, among all the compared auxiliary OOD data based methods in our paper[2][3][4], only DAL reported their result on ImageNet-1K benchmark, while others only evaluated on CIFAR benchmarks and a small subset of ImageNet-1K, e.g. ImageNet-10 that consists of 10 randomly sampled classes, implying their difficulty to implement on large-scaled datasets. Since DAL has shown superior performance compared to other methods on ImageNet benchmark in its paper, we choose it as our comparison baseline.
> 2. Experiment on more architectures: We have tested our method on ImageNet benchmark with ResNet50 architecture. Following your suggestion, we further evaluate our method on transformer based network, e.g. ViT-B-16, on ImageNet benchmark. The training hyper-parameters are the same as our previous setting,  e.g. learning rate 0.0001, epoch 5, and batch size 64, and we use pre-trained model in Pytorch as our initial model. The results are shown in Table 4, where our method achieves lower FPR95 and higher AUROC compared with OE, although in a relatively small range. The potential reason may be due to the limitation of feature dimensions, where ViT-B-16 has 784-dimensional feature space while ImageNet dataset has 1000 categories, makes feature separation difficult because no redundant dimensions can be provided for OOD features. The solution to the feature dimension insufficiency is an interesting direction for our future work, and we have some ideas like inserting feature-ascending linear layers into the network to extend the available dimensions or computing our loss on shallow high-dimensional features where neural collapse phenomenon also occurs[5][6].
>
> Table 4. Results on ViT-B-16 model on ImageNet benchmark. We report the average FPR95/AUROC.
>
> |  | iNaturalist | SUN | Places | Textures | Average | ID Acc |
> | --- | --- | --- | --- | --- | --- | --- |
> | OE | 41.96/90.49 | 65.61/82.30 | 70.20/80.93 | 52.25/85.97 | 57.51/84.92 | 80.05 |
> | DAL | 40.52/90.92 | 65.07/82.39 | 70.17/80.96 | 50.94/86.20 | **56.67**/85.12 | 80.06 |
> | Ours | 40.10/91.06 | 65.58/82.25 | 70.12/81.07 | 51.70/86.13 | 56.88/**85.13** | **80.29** |
>
> Hope the above explanation address your concern and we will conduct more comprehensive experiments in our future work. Also, thanks to your question, motivating us to discuss the requirement/limitation on the feature dimension and its impact on our performance in our final paper.
>
> [2] POEM: Out-of-Distribution Detection with Posterior Sampling, ICML, 2022
>
> [3] How to exploit hyperspherical embeddings for out-of-distribution detection, ICLR, 2023
>
> [4] Energy-based Out-of-distribution Detection, NeurIPS, 2020
>
> [5] Feature Learning in Deep Classifiers through Intermediate Neural Collapse, ICML, 2023
>
> [6] Neural Collapse in the Intermediate Hidden Layers of Classification Neural Networks, arXiv, 2023
>
> **W4-Analysis of Feature Embedding:** Thanks for your valuable suggestion. For embedding visualization, we provide the the 2D and 3D visualization of features of test ID data and unseen OOD data in Figure 2 of our paper. And the feature separability of vanilla, OE-trained, and our model gradually increases, especially, OOD features of our model mostly locate in the dimension that orthogonal to the principal subspace of ID features. Moreover, we evaluate the degree of feature separation in numerical in Table 9 of our paper. The results indicate that our method presents larger feature distance between ID and OOD data. Hope the feature visualization and numerical results meet your suggestion, strengthening the evaluation of our effectiveness.

---

> ### Comment · Reviewer_J1wr · 2024-11-25
>
> Thank you for your response.
>
> - The current model requires a manual selection of techniques depending on whether it is half-trained or fully trained, which is often impractical. Can the proposed method achieve strong performance through a unified process?
>
> - All experiments indicate that Clu loss alone is not effective. It only shows utility when combined with Sep loss and performs well exclusively on the CIFAR-100 dataset. This raises questions about the claimed effectiveness of Clu loss. Could you provide an explanation for this?
>
> - Can the visualizations be improved by using more realistic examples, such as data samples from the benchmark datasets used in the experiments, rather than the simplified two-class examples shown in Fig. 2?

---

> > ### Author Response · Authors · 2024-11-26
> >
> > **Unified-process:**
> >
> > Thanks for your question. We have two stages: Finetune the model with CE loss (Stage I) and Finetune the model with all losses (Stage II). Actually, Stage II is the process we should to discuss for OOD. Stage I is only to guarantee that the model has been well trained on ID data.
> >
> > In our experiments, to follow the setting of previous works, we start from a half-trained model for CIFAR10 and CIFAR100, thereby needs Stage I to ensure network convergence on ID data. More practical setting for OOD is that the model has been well trained, i.e., our experiments on ImageNet, in that case, Stage I is not needed.
> >
> > In practice, the owner of a model certainly knows the statues, whether the model is ready or not. In the case that the owner does not know,  he/she could monitor the training loss, i.e., use Stage  I first and when the accuracy remains unchanged, switch to Stage II. Overall, the training framework of our method can be unified, i.e. Stage I + Stage II.
> >
> > **Effective-of-Clu-loss:**
> >
> > The Clu loss is to enhance Neural Collapse property by aligning ID features to their corresponding class parameters. However, it meanwhile makes all features (include ID and OOD) tend to fall on the subspace spanned by model weights. Therefore, only when combining with Sep loss, it can effectively works to enlarge the discrepancy between ID and OOD features. The total loss is a two-sided force that pulls OOD features into the redundant dimension while confining ID features to the subspace.
> >
> > **Visualization on realistic examples:**
> >
> > We do agree with the reviewer that better visualization will help. But we want to clarify that the current Fig 2 is not a simplified two-class example. The data samples we used to plot our figure is indeed from the CIFAR10 benchmark, where we randomly choose two class of samples, e.g. cat and dog, to visualize their features.
> >
> > **The reason for only showing two class of ID data and the test unseen OOD data is that**: for two class of ID data, their features are actually locate in a two-dimensional space spanned by w_1 and w_2, which we used as dimension reduction matrix and the coordinate system in our figure. (we use linear dimension reduction instead of t-SNE since linear projection can intuitively embody the NC property and reserve most information of ID features)
> >
> > If we visualize the ten class samples in a 2D space, the visualization is not accurate since ID features actually locate in a 10-dimensional space instead of 2D. As two class ID samples occupy two dimension, then the third dimension can be used for visualizing the OOD features, whose coordinate is decided by the principal component of OOD samples.
> >
> > From the 2D and 3D figure in our paper, we can discover that ID features almost locate in the 2D space spanned by model weights, even in the 3D figure, where samples’ coordinate is calculated by projection into the 3-rank dimension reduction matrix consisted of w_1, w_2 and principal component of OOD samples. It reflects that ID features rarely distribute on redundant dimensions, in contrast, OOD features almost locate on redundant dimensions while little component on model weight dimensions.
> >
> > But we do agree with you that showing only two-class examples is not sufficient. **We have included more visualization results for different two classes + OOD examples in Appendix.4 of our revised paper.** In other words, they are different slices for CIFAR-10 dataset.

---

> > > ### Comment · Reviewer_J1wr · 2024-12-01
> > >
> > > I appreciate the response and the new visualization results that were provided. I am still concerned, especially by the following two points.
> > >
> > > - Unified-process
> > >
> > > I am still concerned that the utilization or the proposed method and the requirements of manually setup the two-stage process.
> > >
> > > > "In practice, the owner of a model certainly knows the statues, whether the model is ready or not. In the case that the owner does not know, he/she could monitor the training loss, i.e., use Stage I first and when the accuracy remains unchanged, switch to Stage II. Overall, the training framework of our method can be unified, i.e. Stage I + Stage II."
> > >
> > > This reflects the heuristics and additional manual tuning efforts in the application of the method and somehow further strengthened my concerns.
> > > "the owner of a model certainly knows the statues" -- this is not always true; and the "status" is always blurry and complex for a user.
> > >
> > > - Visualization
> > >
> > > In the visualization figures provided, each figure displays two classes along with OOD samples. Why not visualize samples from all classes together in a single plot?

---

> > > > ### Author Response · Authors · 2024-12-02
> > > >
> > > > Thanks for listing the two specific question.
> > > >
> > > > **1. Unified process**
> > > >
> > > > A unified process is Stage I + Stage II: First, use Stage I (only CE loss) and check its convegence.  Then switch Stage II, i.e., using all losses.
> > > >
> > > > The switch point is related to the judgement on the convergence. An experienced researchers or company may have other methods to judge. More importantly, our method relies on NC and as long as there is NC, our method works well. In other words, more epochs in Stage I will not hurt the OOD performance.  Overall, we do not think finding the switch point is a very big problem, at least, it will not lead our method impractical.
> > > >
> > > > **2. Visualization.**
> > > >
> > > > NC property means that the features of different classes are almost orthogonal to others. Meanwhile, OOD examples are  in another orthogonal direction. Therefore, a 3D image can only well demonstrate two classes ID + OOD.
> > > >
> > > > If we show samples from all classes together in a single image, then the samples  from other classes will be compressed into a single point, which is not a good way of visualization. In fact, the images that displaying two classes along with OOD samples could be viewed as different slices of the ten-dimensional subspace (that samples of ten classes reside in), which is more accurate and clear.
> > > >
> > > > We sincerely hope the above explanations address your concerns.

---

### Author Response · Authors · 2024-11-24
**Looking forward to the feedback**

Dear AC and PC Members,

First and foremost, we sincerely appreciate your time, effort, and insights in reviewing our manuscript.  As the time is coming to the discussion ending, we are eager to engage in further discussions and do our best to address the reviewers' concerns, which might help you with better decisions. Below, we summarize the reviewers comments and our corresponding responses:

**(1) Insufficient experiment on ImageNet benchmark:** we explained the reason for only comparing DAL and OE in detail, and further evaluated our approach on ViT architecture.

**(2) Requirement on feature dimension:** we expect the feature dimension to be larger than the output dimension. But even in dimension insufficiency case, our approach still works, which is experimentally proved.

**(3) Dependence on NC property:**  our approach indeed depends on NC property, but in a relatively relaxed degree, only requiring the ID feature locate in low-dimensional space. We also evaluated our method on imbalanced data where NC property declines, and the improved result demonstrated our effectiveness.

**(4) Concerns about DAL performance:** we expounded the reason for performance discrepancy, and listed the results reported in other papers to prove the validity of our results.

 **(5) Training strategy choice:** we explained the choice criteria and highlighted that our approach is not sensitive to the strategy choice.

 **(6) Modest improvement:** we clarified the improvement scale of our method is not modest with comparison to the result reported in other papers, and highlighted that our method is applicable to large-scaled datasets.

We have replied to other concerns with detailed, point-by-point responses. We sincerely hope the reviewers could provide further feedback, and look forward to continuing discussions with you.

Best regards,

The Anonymous Author(s) of Paper9153

---

### Meta-Review · Area_Chair_Vnex · 2024-12-23

**Metareview:**

The authors propose to include two additional terms to standard OE (outlier exposure) loss function for improved out of distribution (OOD) detection-- the first term is to ensure extremely compact clustering of in-distribution samples while the second one ensures that the OOD features lie on the subspace orthogonal to the subspace spanned by the features (weight vectors) of ID classes. This new loss has been shown to perform better in OOD detection tasks over several benchmarks.

The approach seems to be simple and intuitive, and the results are promising. I found the dependency of the new loss function on too many hyper-parameters a bit concerning however the authors showed that the method is not much sensitive to the choice of hyper-parameter.

I initially considered this work to be a borderline but I must admit that the comprehensive responses from the authors to the extremely relevant questions by the reviewers did not only convince most of the reviewers but also to me that this work would be a good fit to ICLR. However, the authors must incorporate all the comments by the reviewers in their revised draft. They all make sense and will surely increase the visibility of this work.

**Additional Comments On Reviewer Discussion:**

The reviewers have done a great job in asking the right questions with proper engagement throughout which, as I can see, have surely improved the quality and understanding of this work. Few important comments from the reviewers that the authors should pay proper attention to in their revised draft are:

- clarify two stages of training to avoid any confusion
- talk about data variability and resulting performance discrepancies (sampling strategies etc)
- discuss (briefly) the similarity/dissimilarity with kNN
- include additional experiments in the main paper

---

### Decision · Program_Chairs · 2025-01-22

Accept (Poster)